# Machine learning derived retinal pigment score from ophthalmic imaging shows ethnicity is not biology

Anand E. Rajesh[1,2,37], Abraham Olvera-Barrios [3,37], Alasdair N. Warwick[3,4], Yue Wu[1,2], Kelsey V. Stuart [3], Mahantesh I. Biradar [3], Chuin Ying Ung[5], Anthony P. Khawaja [3,6], Robert Luben [3,6], Paul J. Foster [3], Charles R. Cleland[7,8], William U. Makupa[8], Alastair K. Denniston [9], Matthew J. Burton[3,7], Andrew Bastawrous[8,10], Pearse A. Keane [3], Mark A. Chia[3], Angus W. Turner [11], Cecilia S. Lee [1,2], Adnan Tufail [3], Aaron Y. Lee [1,2], Catherine Egan [3] ✉ & UK Biobank Eye and Vision Consortium*

Few metrics exist to describe phenotypic diversity within ophthalmic imaging datasets, with researchers often using ethnicity as a surrogate marker for biological variability. We derived a continuous, measured metric, the retinal pigment score (RPS), that quantifies the degree of pigmentation from a colour fundus photograph of the eye. RPS was validated using two large epidemiological studies with demographic and genetic data (UK Biobank and EPIC-Norfolk Study) and reproduced in a Tanzanian, an Australian, and a Chinese dataset. A genome-wide association study (GWAS) of RPS from UK Biobank identified 20 loci with known associations with skin, iris and hair pigmentation, of which eight were replicated in the EPIC-Norfolk cohort. There was a strong association between RPS and ethnicity, however, there was substantial overlap between each ethnicity and the respective distributions of RPS scores. RPS decouples traditional demographic variables from clinical imaging characteristics. RPS may serve as a useful metric to quantify the diversity of the training, validation, and testing datasets used in the development of AI algorithms to ensure adequate inclusion and explainability of the model performance, critical in evaluating all currently deployed AI models. The code to derive RPS is publicly available at: https://github.com/uw-biomedical-ml/retinal-pigmentation-score.

Retinal diseases are a significant global cause of vision loss, but not all populations are affected equally. In 2020, there were estimated to be 103.1 million adults worldwide with diabetic retinopathy (DR) and 196 million people with age-related macular degeneration (AMD)[1]. Studies have found DR prevalence is highest in Africa (35.90%), then North America and the Caribbean (33.30%). In contrast, AMD has a significantly higher prevalence in people of European than in those of Asian or African ancestry[2,3]. In response to the overwhelming global

[1]Department of Ophthalmology, University of Washington, Seattle, WA, USA. [2]The Roger and Angie Karalis Johnson Retina Center, Seattle, WA, USA. [3]NIHR Biomedical Research Centre, Moorfields Eye Hospital NHS Foundation Trust & University College London Institute of Ophthalmology, London, UK. [4]University College London Institute of Cardiovascular Science, London, UK. [5]Guy's and St Thomas' NHS Foundation Trust, London, UK. [6]MRC Epidemiology Unit, University of Cambridge, Cambridge, UK. [7]International Centre for Eye Health, Faculty of Infectious and Tropical Diseases, London School of Hygiene & Tropical Medicine, London, UK. [8]Eye Department, Kilimanjaro Christian Medical Centre, Moshi, United Republic of Tanzania. [9]NIHR Birmingham Biomedical Research Centre, Birmingham, UK. [10]PEEK Vision, Berkhamsted, UK. [11]Lions Eye Institute, University of Western Australia, Nedlands, WA, Australia. [37]These authors contributed equally: Anand E. Rajesh, Abraham Olvera-Barrios. *A list of authors and their affiliations appears at the end of the paper. ✉e-mail: cathy.egan@nhs.net

burden of disease, many artificial intelligence (AI) algorithms have been developed to enable more efficient care delivery. These AI algorithms have been widely published, and several of them are already in clinical practice for providing automated diagnoses of diseases such as DR, AMD, and glaucoma[4–8].

The success of AI algorithms in ophthalmology is partly due to the availability of large imaging datasets that have been collected from routine clinical practice[9]. Describing the demographic characteristics of these large datasets used for training is critical for understanding algorithm generalisability, limitations and overall performance. Despite this need, less than 20% of publicly available retinal imaging datasets contain patient characteristics such as age, sex, or ethnicity[10]. Studies that compare model performance across different populations are limited due to methodological limitations and unlabelled data, making bias evaluation challenging[4,11,12].

Previous studies assessing bias in general AI algorithms have found that the performance of image-based algorithms is often worse among people with a greater degree of skin pigmentation, for instance, in skin cancer classification, facial recognition and object detection[13–15]. In these studies, image pixel values are measured and converted into a categorical pigmentation scale by human labellers or an algorithm. These categories are then used to estimate the relative performance of algorithms within subcategories of pigmentation, either in the presence or absence of additional demographic data. Additionally, these scales allow for the decoupling of race/ethnicity from the biological differences in skin pigmentation[16].

In the eye, melanin is present in the uvea (iris, retina, and choroid) and is responsible for blue or brown iris colour and retinal pigmentation[17–22]. We aimed to develop a continuous scale called the Retinal Pigment Score (RPS) to quantify the background pigmentation of retinal colour fundus photographs captured in UK Biobank. We then sought to validate the RPS by comparing against the self-reported ethnicity of the UK Biobank participants and by performing both genome-wide (GWAS) and phenome-wide (PheWAS) association studies. Mechanistic insight was attained through gene prioritisation and

functional annotation whilst causal associations with clinically relevant outcomes were tested using Mendelian randomisation. We validated our results with a replication GWAS study in an independent cohort (European Prospective Investigation into Cancer and Nutrition [EPIC]--Norfolk) and reproduced the RPS method in three non-white populations (Tanzanian, Australian and Chinese datasets).

## Results

### Retinal Pigment Score

We designed an algorithm to extract the background retinal pigmentation from colour fundus photographs. Briefly, the algorithm uses published open-access deep learning models[23] to identify and exclude ungradable images, create a mask of the tissue that excludes the retinal vasculature and optic nerve, then finds the average chromaticity of this background tissue. Chromaticity in this study is the quality of colour independent of illumination measured in the CIELAB colour space[24]. The chromaticity is then converted into a single continuous metric: the RPS (Fig. 1, Methods Retinal Pigmentation Score). A greater RPS correlates with a greater degree of pigmentation in the retina. A total of 135,592 colour fundus photographs (67,982 right eyes, 67,610 left eyes) from 68,504 participants in the UK Biobank study were available for analysis. From these, 74,851 images (40,329 right eyes, 34,388 left eyes) from 44,320 participants (55% female) were deemed gradable by our pipeline and included in the analysis (Supplementary Table 1). A previous study of this dataset deemed only 11% of the images as "Good" quality when assessed by human graders. Additionally, this study described a similar imbalance in laterality[25]. Supplementary Fig. 1 shows the percentage area of the image identified as vessels and optic disc was comparable across ethnic groups. Moreover, the approximate area of the optic nerve head would fall in line with previous reports by ethnicity[26]. Additionally, small differences in the size of the segmentation masks of the vessels and optic disc have a minimal effect on the RPS, as shown in Supplementary Fig. 2. The patient-level characteristics are summarised in Supplementary Table 2. The median age was 56 years (Interquartile Range[IQR]: 49-63) and 92% (40,704/44,320) of

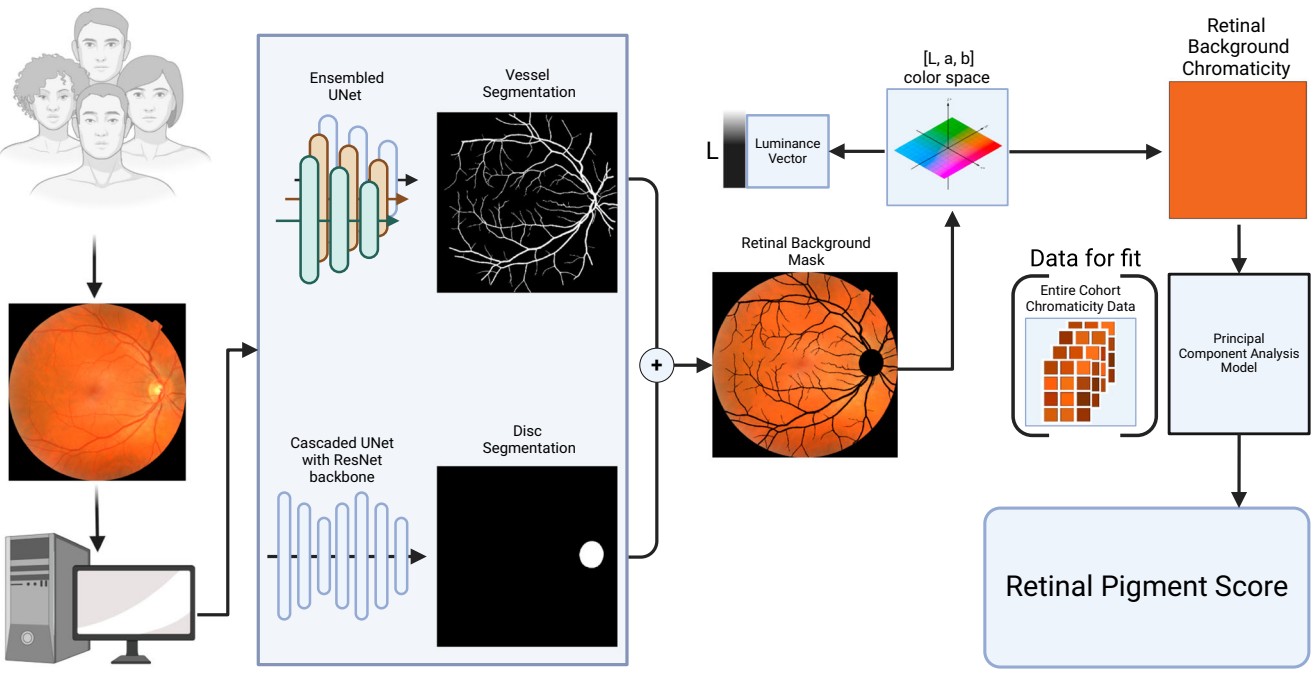

**Fig. 1 | Schematic showing the method to generate the retinal pigmentation score (RPS) from a colour fundus image.** Input images are fed into the deep learning algorithm to generate segmentation masks. These are added together to make a retinal background mask, which is then transformed into L,a,b colorspace. The chromaticity vectors are then extracted and transformed by the principal component analysis model to create the RPS. Created with Biorender.com.

participants self-described their ethnicity as white. The proportional eigenvalues of the Principal Component Analysis used to normalise the RPS of the first two dimensions were 0.899 and 0.101. The median (IQR) RPS was −0.82 (−9.89, 10.39). Each ethnic group had images in every quintile of the RPS range, except for Chinese participants in the lowest (less pigmented) quintile (Fig. 2).

Our RPS method was reproduced in three independent non-white populations from cohorts outside of the UK (described here by their geographical location as "Tanzanian, Australian, and Chinese" datasets. Self-described ethnicity data was not available for the Tanzanian and Chinese datasets and was described as Indigenous Australian for the Australian dataset). To aid in comparison, we elected to fit the RPS model trained on the UK Biobank to these datasets to create RPS with a unified scale. A total of 1150 colour fundus photographs of 675 eyes from 348 people were included in the Tanzanian dataset, 715 colour fundus photographs of 715 eyes from 439 people were included in the Australian dataset, and 2088 colour fundus photographs of 2088 eyes from 1431 people were included from the Chinese dataset. The Chinese dataset was from the publicly available Ocular Disease Intelligent Recognition (ODIR) dataset[27]. Supplementary Table 1 provides details on the number of images analysed, included, and deemed as inadequate quality, hence excluded by the pipeline. The median RPS were 40.24 (IQR: 36.38–44.94) in the Tanzanian dataset, 27.92 (18.67–35.17) in the Australian dataset, and 29.79 (21.94–35.41) in the Chinese dataset. Figure 2 shows the RPS distribution from Tanzanian, Australian, and Chinese datasets in relation to the RPS distribution of self-described ethnicity groups from the UK Biobank.

### Retinal pigment score reliability
We assessed RPS reliability in an independent dataset of 27 eyes from 24 patients with two images per eye. The one-way consistency intraclass correlation coefficient (ICC) was 0.920 (95% CI 0.834, 0.963, $p$-value < 0.001).

Among 30,407 participants in the UK Biobank that had available imaging, the mean and standard deviation of the difference in RPS between the right and left eye was −1.36 (8.30) and the one-way ICC was 0.788 (95% CI: 0.784, 0.792). In contrast, the use of the L, a, b vectors to calculate RPS among the same group of participants yielded a mean score of −2.40 (10.70) and a one-way ICC of 0.757 (95% CI: 0.753, 0.762).

### Associations of RPS with clinical variables
We first examined associations of mean RPS (average score between right and left eyes per participant) with sociodemographic and clinical variables. Supplementary Fig. 3 shows the association of RPS with the covariates of interest (deciles of continuous variables) adjusted for age, sex, and UK Biobank centre. Non-white self-described ethnicities were associated with increased RPS when compared with white individuals. A positive graded association was observed with increased skin pigmentation, hair pigmentation, and deprivation. There was an inverse linear association between RPS and height.

Next, the associations were tested with multivariable linear regression adjusting for age, sex, height, self-described ethnicity, self-described hair and skin colour, Townsend deprivation index (TDI), refractive status, and UK Biobank assessment centre (Supplementary Table 3). The RPS was modelled as a z-score. Coefficients represent the standard deviation (SD) change in RPS per specified increase in covariates or the standardised difference between groups. Formal variance inflation factor testing on the final model with adjusted generalised standard error inflation factors showed no strong collinearity (Supplementary Table 4) When compared with very fair skin colour, darker skin tones showed a graded increase in RPS (p for linear trend $2.5 \times 10^{-231}$). People with black skin colour showed a 0.83 SD increase in RPS (95%CI 0.72, 0.94; $p$ $5.7 \times 10^{-48}$) when compared with

people of very fair skin colour. Similarly, when compared with people with blonde hair, darker hair colours showed a graded positive association with RPS ($p$ for linear trend $2.2 \times 10^{-155}$). People with black hair colour showed a 0.53 SD increase in RPS when compared with people with blonde hair colour ($p$ $4.8 \times 10^{-122}$).

There was a strong association of ethnicity with RPS. When compared with white individuals, Chinese (1.49, [1.35, 1.62]; $p$ $4.6 \times 10^{-103}$) and black (1.15, [1.07, 1.23]; $p$ $1.6 \times 10^{-158}$) people showed the largest effect sizes. However, within ethnic groups, there was a wide spread of overlapping RPS values (Fig. 2). Every 5-year rise in age was associated with a small 0.02 SD increase in RPS ($p$ $1.3 \times 10^{-8}$), and every 5 cm increase in height conferred a small −0.02 SD change in RPS ($p$ $3.6 \times 10^{-8}$). However, sensitivity analyses with stratified linear regression models across the three main ethnic groups showed an association in different direction for age in white ethnic groups when compared with models from Black and Asian ethnic groups (Supplementary Table 5). The association with height remained significant and in the same direction for white and Asian ethnic group models, and was not significant for Black ethnic groups. Supplementary Fig. 4 shows mean RPS adjusted for sex, and UK Biobank centre by deciles of age and height for the three main ethnic groups. A non-linear association was evidenced for refractive status. A higher RPS was observed in people with emmetropia (0.16 [95%CI 0.11, 0.20]; $p$ $1.1 \times 10^{-12}$), and hyperopia (0.11, [0.06, 0.15]; $p$ $1.1 \times 10^{-6}$) when compared with people with high myopia. The most deprived TDI quintile showed a 0.06 SD increase in RPS when compared with the least deprived TDI ($p$ for linear trend $3 \times 10^{-4}$). Townsend Deprivation Index showed, however, an association in a different direction in sensitivity analysis in the white ethnic group model when compared with the Black ethnic group model (Supplementary Table 5). Sex was not associated with RPS.

### Genome-wide association study discovery analysis
A genome-wide association study (GWAS) was performed to assess potential associations with standardised mean RPS (average score between right and left eyes per participant); robust associations would potentially support the biological plausibility of the metric. The discovery analysis included 37,067 individuals of European ancestry from the UK Biobank cohort. The genomic inflation factor was 1.071, and the linkage disequilibrium score regression intercept was 1.013 with a ratio of 0.09. Conditional analysis identified 20 independent autosomal genomic loci reaching genome-wide significance ($p < 5 \times 10^{-8}$), the majority of which (17/20) have previously been shown to associate with hair, skin and/or iris colour (Table 1, Fig. 3).

Positional and expression quantitative trait locus mapping in retina, skin and dermal fibroblasts were performed to identify candidate causal genes at each independent risk locus. This produced a set of 100 prioritised genes (Supplementary Table 6), which were then annotated in biological context. A number of these had existing entries in the GWAS Catalog[28], with enrichment for traits including hair, eye and skin colour, as well as various skin malignancies (Supplementary Fig. 5). Enrichment for several Gene Ontology entries was also apparent, especially those related to melanin and pigmentation biological processes (Supplementary Fig. 6).

### Genome-wide association study replication analysis
A replication GWAS was conducted in the independent EPIC Norfolk Eye Study cohort, which is a predominantly white (99.7%), longitudinal cohort from Norfolk, England[29]. This replication analysis included 4273 individuals of European ancestry. Due to differences in genotyping platforms and imputation methods, three of the lead variants highlighted in the discovery GWAS were either unavailable or did not pass quality control in the replication dataset (rs173273, rs762948237, and rs766338951). Replication was therefore assessed for 17 out of the 20 lead variants.

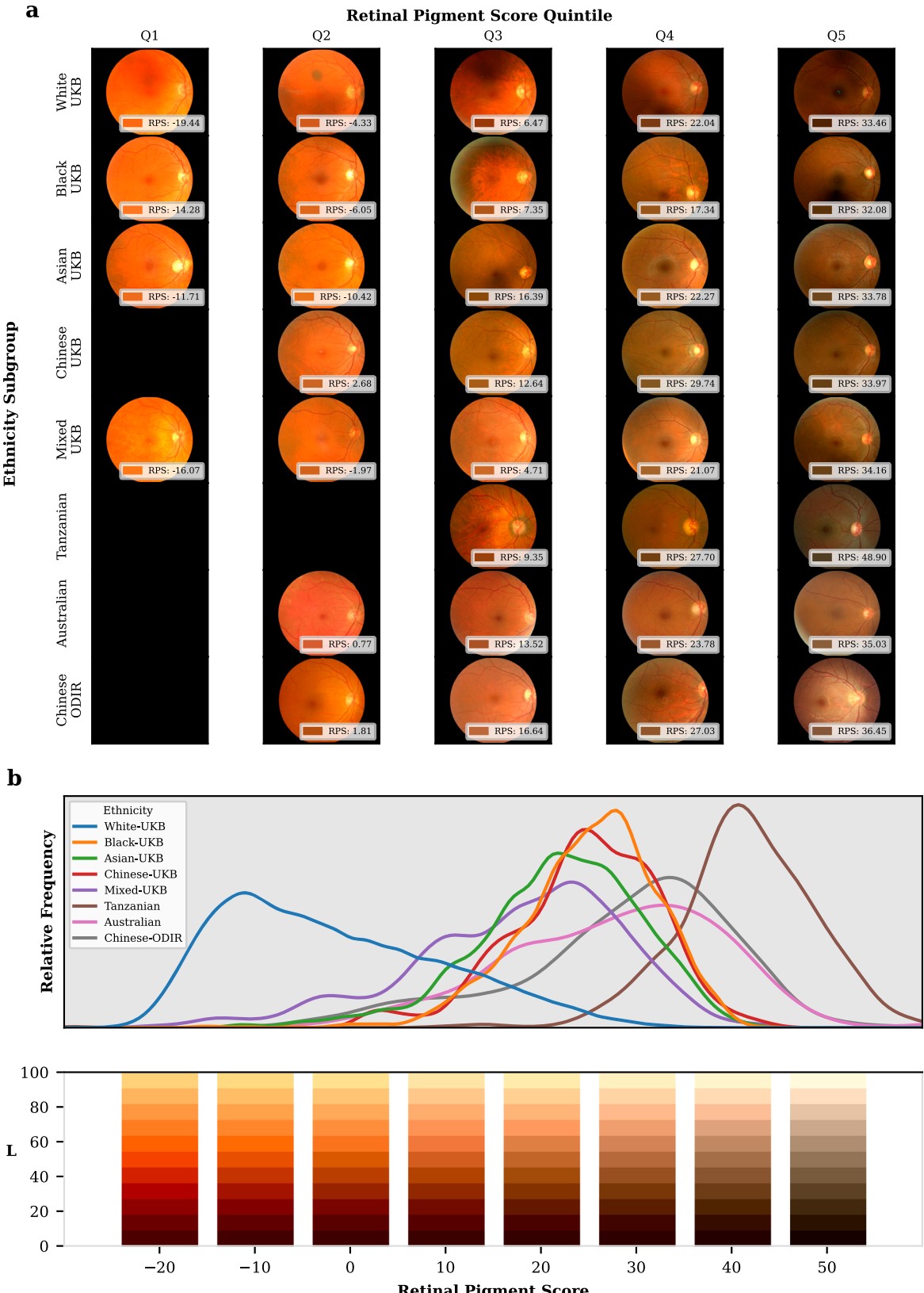

**Fig. 2 | Representative fundus photos with associated RPS. a** Randomly sampled colour fundus photographs from each UK Biobank self-reported ethnicity and from the Tanzanian, Australian, and Chinese (ODIR) datasets, sorted by quintiles of retinal pigment score (RPS) across the entire distribution of RPS for the UK Biobank cohort. The RGB colour of the pixel value that is converted into RPS as well as the RPS is shown at the bottom of each fundus photograph. Black spaces represent when there are no suitable images within the respective ethnicity subgroup and quintile

**b** Normalised kernel density estimation plot of the distribution of RPS for all participants grouped by self-reported ethnicity as reported in the UK Biobank as well as the Tanzanian, Australian, and Chinese (ODIR) datasets. Relative frequencies are normalised so the area under each curve is equal for each ethnicity subgroup. The subpanel consists of examples where for a given RPS and the a,b values in the CIELAB colour space are constant but the L vector changes. The x-axis is shared in both subpanels. Source data are provided as a Source Data file.

**Table 1 | Genome-wide significant associations with retinal pigment score in the UK Biobank cohort**

| Rs identifier | chr:pos [hg19] | EA/OA (EAF) | Beta (95% CI) | P | Nearest gene | Hair colour | Skin colour | Eye colour | Replicated |
|---|---|---|---|---|---|---|---|---|---|
| rs6670870 | 1:205155177 | A/T (0.76) | −0.09 (−0.11; −0.08) | 8.7E−36 | DSTYK | Yes | | | |
| rs173273 | 1:212446689 | G/T (0.41) | 0.04 (0.02; 0.05) | 2.9E−08 | PPP2R5A | Yes | Yes | | |
| rs762948237 | 3:129178587 | TCTTC/T (0.87) | 0.05 (0.03; 0.07) | 2.3E−08 | IFT122 | | | | |
| rs16891982 | 5:33951693 | C/G (0.02) | 0.52 (0.48; 0.56) | 1.5E−135 | SLC45A2 | Yes | Yes | Yes | Yes |
| rs12203592 | 6:396321 | C/T (0.79) | 0.13 (0.11; 0.14) | 2.4E−59 | IRF4 | Yes | Yes | Yes | Yes |
| rs62425803 | 6:134330249 | G/A (0.81) | 0.05 (0.04; 0.07) | 1.4E−11 | TCF21 | Yes | | | |
| rs117756744 | 7:100277212 | G/A (0.98) | 0.18 (0.14; 0.23) | 5.4E−17 | GNB2 | Yes | Yes | | Yes |
| rs1325117 | 9:12613472 | G/A (0.36) | 0.06 (0.05; 0.07) | 3.0E−19 | TYRP1;LURAP1L | Yes | Yes | | Yes |
| rs11023814 | 11:16007053 | C/G (0.43) | 0.04 (0.03; 0.06) | 2.2E−12 | SOX6 | | Yes | | Yes |
| rs150527451 | 11:68817897 | G/A (0.89) | 0.16 (0.14; 0.18) | 1.7E−53 | TPCN2 | Yes | Yes | | Yes |
| rs1060435 | 11:68855595 | A/G (0.59) | 0.07 (0.05; 0.08) | 1.1E−24 | TPCN2 | Yes | Yes | Yes | Yes |
| rs747572 | 11:87885082 | A/G (0.63) | 0.05 (0.04; 0.06) | 4.9E−14 | CTSC | Yes | Yes | | Yes |
| rs1126809 | 11:89017961 | G/A (0.7) | 0.07 (0.06; 0.09) | 5.0E−27 | TYR | Yes | Yes | Yes | Yes |
| rs4762973 | 12:20710145 | A/G (0.75) | 0.06 (0.04; 0.07) | 7.8E−15 | PDE3A | | | | |
| rs10771034 | 12:23979199 | T/A (0.45) | −0.04 (−0.06; −0.03) | 1.1E−12 | SOX5 | Yes | Yes | | Yes |
| rs766338951 | 13:95169060 | CT/C (0.69) | 0.08 (0.06; 0.09) | 4.6E−30 | DCT | Yes | Yes | | |
| rs1800407 | 15:28230318 | C/T (0.91) | 0.11 (0.09; 0.13) | 1.8E−23 | OCA2 | Yes | Yes | Yes | Yes |
| rs12913832 | 15:28365618 | A/G (0.22) | 0.44 (0.43; 0.46) | 0.0E+00 | HERC2 | Yes | Yes | Yes | Yes |
| rs7220155 | 17:79606020 | C/T (0.62) | −0.06 (-0.07; −0.05) | 9.2E−22 | TSPAN10 | Yes | Yes | | Yes |
| rs1785433 | 21:44783282 | A/G (0.65) | −0.04 (-0.05; −0.02) | 1.3E−08 | SIK1 | | | | Yes |

Variants that met the replication threshold of two-sided p < 0.05 measured with the chi-square test statistic in the EPIC-Norfolk replication GWAS are indicated in the 'Replicated' column. There are 3 columns to indicate which variants have previously been shown to be associated with hair, skin or iris colour.

The direction of effect was concordant for all 17 variants and highly correlated with estimates from the discovery analysis (Pearson's rho = 0.986 [95% CI: 0.961, 0.995]) (Fig. 4). Of the 17 variants, 15 variants were significant at $p < 0.05$, 8 remained significant after adjusting for multiple testing ($p < 0.05/17$), and 2 achieved genome-wide significance (Supplementary Table 7).

**Phenome-wide association study**
A phenome-wide association study (PheWAS) was performed within the discovery UK Biobank GWAS sub-cohort ($n = 37,067$) to assess potential associations between standardised mean RPS (average score between right and left eyes per participant) with 308 diseases. After correction for multiple testing ($p < 0.05/308$), significant associations were observed for higher RPS (indicating more pigmentation of the retina) with decreased odds of 'Actinic keratosis' (OR: 0.87, 95% CI: [0.81, 0.93]), 'Primary Malignancy - Other Skin and subcutaneous tissue' (0.90 [0.86, 0.94]), and 'Migraine' (0.91, [0.87, 0.95]). Higher RPS was associated with a greater risk of chronic obstructive pulmonary disease (COPD); (1.11, [1.05, 1.16]). A further 26 diseases were significantly associated at $p < 0.05$, including decreased odds for 'Primary Malignancy - Malignant Melanoma' ($p = 0.02$) (Supplementary Fig. 7, Supplementary Table 8).

**Mendelian randomisation**
Two-sample Mendelian randomisation (MR) analyses were performed to probe potential causal relationships between genetically predicted retinal pigmentation with outcomes of particular interest, as highlighted by the PheWAS analysis, using outcome summary statistics from FinnGen[30] (Supplementary Table 9). 13 variants were included in the instrumental variable following clumping of the 20 conditionally independent lead variants from the discovery RPS GWAS, and harmonisation with FinnGen (see Methods). All variants had an F-statistic >10 (mean 139.9). Inverse-variance weighted (IVW) MR estimates provided evidence for protective causal effects on actinic keratosis (OR 0.44 per SD RPS [0.24, 0.83]; $P = 0.01$), basal cell carcinoma of the skin (OR 0.59 per SD RPS [0.38, 0.92]; $P = 0.02$), squamous cell carcinoma

of the skin (OR 0.38 per SD RPS [0.20, 0.73]; $P = 0.003$), non-melanoma skin cancer (OR 0.40 per SD RPS [0.22, 0.73]; $P = 0.03$) and malignant melanoma of the skin (OR 0.60 per SD RPS [0.38, 0.94]; $P = 0.003$). These findings were supported by weighted median, weighted mode, and MR-Egger sensitivity analyses (Supplementary Table 10, Supplementary Fig. 8–12). Despite the presence of global heterogeneity for the MR instrument (Cochran's Q statistic $P < 0.001$), the MR-Egger intercept test did not indicate average directional pleiotropy ($P > 0.05$) (Supplementary Table 11). There was no evidence for a causal relationship with either COPD or migraine (Supplementary Table 10, Supplementary Figs. 13, 14).

## Discussion
We introduce a metric, the RPS, which quantifies the background pigmentation of the retina from colour fundus photographs along a continuous scale and is strongly associated with genetic variants linked to human skin, eye, and hair phenotypes with replication in an additional cohort. We studied datasets derived from people living in the UK, including more than 3000 people who describe their own ethnicity as not white, and from people with diabetes living in Tanzania (with a single self-reported ethnicity), people living in Australia who describe their ethnicity as indigenous Australian or Aboriginal and Torres Strait Islander, and more than 1,400 people living in China. The RPS captures the biological variability of retinal colour without recourse to the distinct, social and political constructs of race and ethnicity. In our study, there is a significant overlap in the distribution of RPS among a variety of ethnic groups and a wide range of RPS within each ethnicity (Fig. 2). Many self-described ethnicities have RPS that fall within each quintile of the RPS distribution, a feature that makes it impossible to determine ethnicity from RPS alone.

Pigmentation is found everywhere in the body and is often studied in relation to skin tone. In the field of dermatology, many scales describe the degree of pigmentation in the skin. The Fitzpatrick classification of skin types (FST), developed in 1975, categorises skin colour into 6 types based on the skin's reaction to UV radiation exposure,

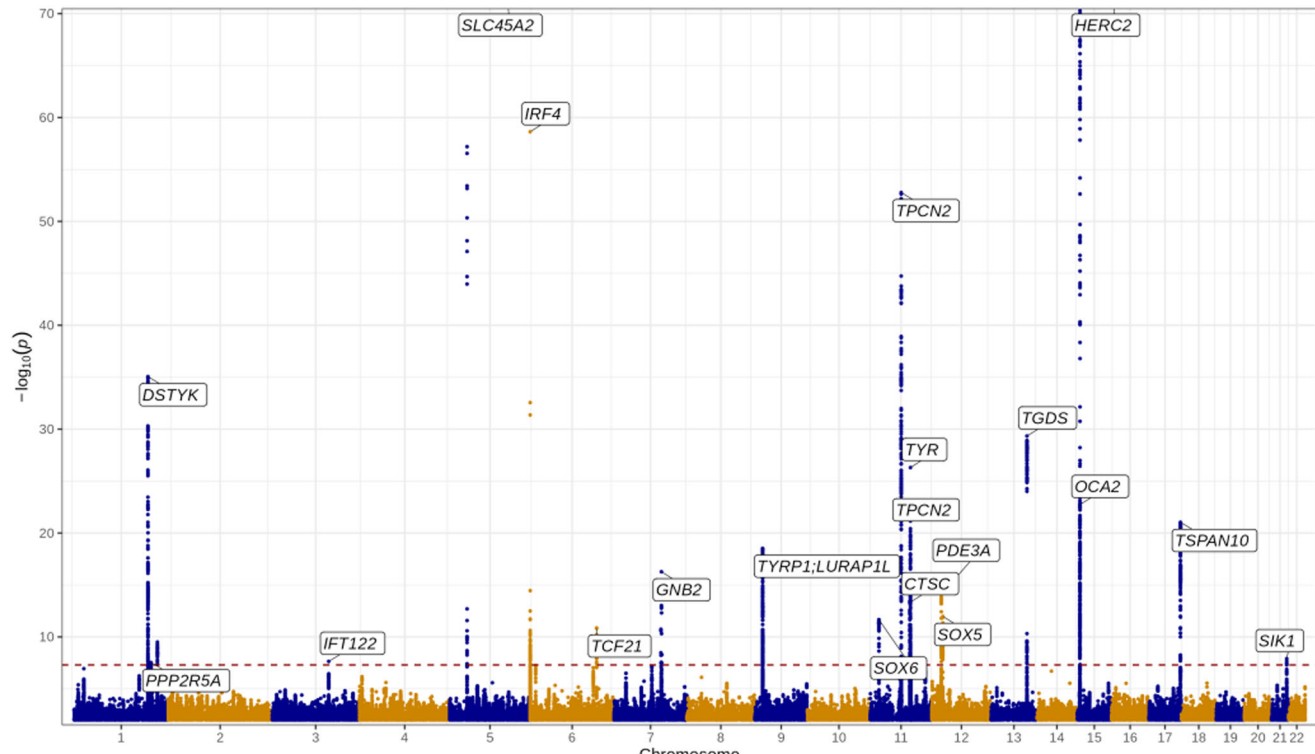

**Fig. 3 | Manhattan plot of GWAS results from the discovery cohort (UKBiobank, *n* = 37067).** The *Y*-axis represents the two-sided *p*-values from the linear mixed effects model. Lead variants identified by GCTA-COJO are annotated with the nearest gene. Points are truncated at −log10(*p*) = 70 for clarity. The dashed red line indicates genome-wide significance ($p = 5 \times 10^{-8}$) which is adjusted for multiple comparisons and the *p*-values are two-sided and calculated with the z-statistic. Source data are provided as a Source Data file.

from type I (fair skin, always burns, never tans) to type VI (deeply pigmented, never burns)[31,32]. This scale has been adopted in the field of computer science to describe the diversity in imaging datasets, and expose underlying biases within AI algorithms. Studies have found that facial recognition AI performs worse on individuals with darker skin colour measured by the FST scale[14] and object detection software is worse at detecting darker skinned pedestrians by FST scale from images of street traffic[13]. This led to the Google Ethical AI team recommending that all computer vision models report their performance across a range of FST scales[33]. Recent work to widen the range of skin tones has also been adopted as an industry standard[34].

There is some evidence that retinal colour affects model performance. A deep learning model trained to predict age-related macular degeneration (AMD) found that patients with the minor allele at the rs12913832/*HERC2* locus, associated with retinal pigmentation in our study, were more likely to have false positives for age-related macular degeneration[35]. Another study postulated that drusen, a pathological feature of AMD, may be more noticeable against a darker fundus background[36]. Hirsch et al. showed saturation values from retinal oximetry vary according to retinal pigmentation[37,38]. We found associations between the RPS and multiple genetic loci previously associated with skin, hair and iris colour, providing strong biological evidence that the RPS does indeed reflect the degree of retinal pigmentation. Of the 20 genome-wide significant loci identified by conditional analysis in the discovery GWAS analysis, 17 had pre-existing evidence for being associated with hair, skin or iris pigmentation, including 3 that are known to be associated with oculocutaneous albinism (*TYR*, *OCA2* and *TYRP1*)[39]. Furthermore, despite differences in study populations and cameras, we observed robust replication for these loci in the EPIC-Norfolk cohort and a strong correlation between beta coefficients in the two cohorts. This suggests that despite a range of input

data characteristics, the RPS is still estimating retinal pigmentation. Post-GWAS analyses for a set of 100 prioritised causal genes demonstrated enrichment for various melanin and pigmentation pathways, as well as enrichment for pigmentation-related traits in the GWAS Catalog[28].

The two most significantly associated loci in the discovery GWAS analysis were at *HERC2* (rs12913832), and *SLC45A2* (rs16891982). These also reached genome-wide significance in the replication analysis. The former is known to influence melanin production via effects on *OCA2* expression, and iris colour[40,41] while rs16891982 is a missense mutation in the *SLC45A2* gene[42]. rs12913832 modulates human pigmentation by affecting chromatin-loop formation between a long-range enhancer and the *OCA2* promoter, leading to decreased expression of OCA2 and lighter pigmentation[40]. This variant is strongly associated with brown iris colour in European populations[41]. The *SLC45A2* gene encodes a membrane protein involved in the transport of solutes including tyrosine (a precursor to melanin synthesis), which is implicated in the regulation of skin, hair and iris colour[43–45]. rs16891982 encodes a missense mutation in *SCL45A2*, and has been associated with skin pigmentation as well as a strong association with risk for cutaneous malignant melanoma[46].

Interestingly, the lead variants at *PDE3A*, *SIK1* and *IFT122* have not been previously associated with skin, hair or iris pigmentation, thus these new variants may be specifically related to retinal pigmentation. *PDE3A* has been previously associated with arteriolar tortuosity[47], *SIK1* with the regulation of circadian rhythms[48], and in vitro work has implicated alternative splicing for *IFT122* to play a role in PRPF31 retinitis pigmentosa pathogenesis[49].

The PheWAS analysis suggested that some diseases associated with skin pigmentation were also associated with retinal pigmentation. Both actinic keratosis and cutaneous malignancy were inversely

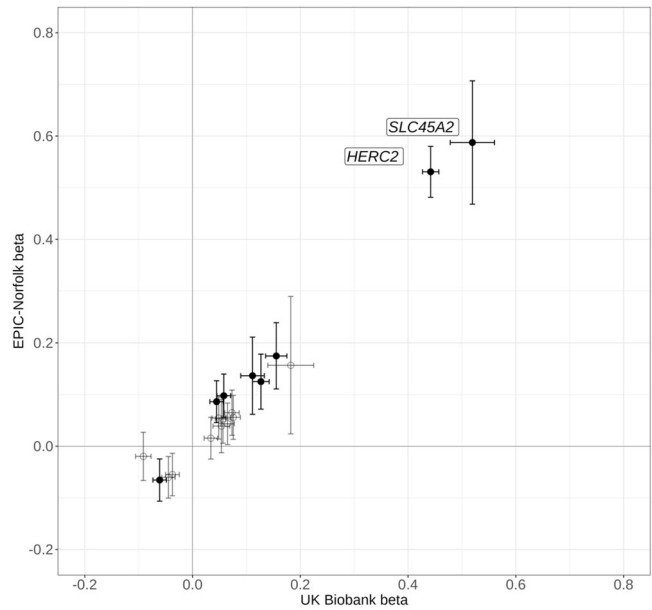

Meets Bonferonni-adjusted replication significance threshold: ○ No ● Yes

**Fig. 4 | Comparison of betas for lead variants identified from the discovery and replication cohort.** Comparison of betas expressed as change in standard deviation of mean RPS for lead variants identified from the discovery (UK Biobank, $n = 37067$) genome-wide association study (GWAS) with their corresponding betas in the replication (EPIC-Norfolk, $n = 4273$) analysis, with 95% confidence intervals. Betas in the cohort were calculated using a generalised linear mixed model, adjusting for age, sex and the first ten principal components. *P*-values are two-sided, calculated from the z-statistic and corrected for multiple comparisons. Variants meeting the Bonferroni-adjusted replication significance threshold ($p = 0.05$/ variants) in the EPIC-Norfolk GWAS are shaded black. The nearest gene is annotated for variants achieving genome-wide significance. Source data are provided as a Source Data file.

associated with increased RPS at phenome-wide significance. Malignant melanoma was also inversely associated with increased RPS, albeit only nominally, possibly due to the limited number of participants (n = 643) with malignant melanoma. MR analyses furthermore provided evidence that genetic predisposition to increased retinal pigmentation is causally protective for skin malignancies, including malignant melanoma. Migraine and COPD also showed phenome-wide significant associations with RPS. The observational association with migraine is consistent with a recent study showing an increased risk of migraine with lightly pigmented skin[50]. Similarly, the effect direction of the association of RPS with COPD and allergic rhinitis is concordant with previous literature postulating skin pigmentation and vitamin D relationships[51–53]. However, further investigation with MR did not yield any evidence for causality between retinal pigmentation with either migraine or COPD.

The RPS may be utilised for several indications in the future. Reporting the RPS range for any AI training datasets could offer an immediate description of the phenotypic diversity within the datasets. Training or test sets could incorporate the RPS in the descriptive summaries along with other important biological and demographic covariates. The RPS could be used as a standard metric in evaluating differences in AI algorithm performance across a diverse set of input images, similar to how FST has been used in dermatology. This would help ensure that real-world biological variability in health and disease remains included in the generation of new algorithms. This has implications not just for ophthalmic diseases, but also for other diseases using retinal imaging biomarkers, such as Alzheimer's disease[54,55] and cardiovascular disease[56,57]. With

the genetic associations demonstrated in this study, the RPS may be used to study retinal pigmentation itself in relation to normal eye and brain development, and a myriad of ophthalmic and systemic diseases.

There are several limitations in this study. Both the UK Biobank and EPIC-Norfolk participants are predominantly self-reported white and European. The UK Biobank has a higher proportion of non-white participants compared to EPIC-Norfolk cohort, but this number comprises only 7.5% in the UK Biobank. Fortunately, this number still equates to over 3000 people reporting non-white ethnicity. We also showed that the RPS generated separately in an ethnically distinct Tanzanian, Australian, and Chinese dataset is feasible when fit to the UK Biobank RPS. It is our hope that the open-source availability of the RPS method will allow this technique to be applied to other datasets to address this limitation. Secondly, RPS is currently dataset-specific, so that absolute RPS values from different cohorts cannot be directly compared if they are not fit to the same RPS scale. In this work, we elected to fit the Chinese, Australian and Tanzanian datasets to the UK Biobank RPS scale to aid in comparison. This may be resolved with standardisation of the metric between camera types, using device-specific raw RGB values, and is the subject of future work. Thirdly, large datasets that link genetic data with retinal images are only available in limited geographic areas in the world. Finally, the performance of the RPS in retinal disease states will need to be assessed in future work.

In conclusion, the RPS is a continuous metric of retinal pigmentation directly derived from a non-invasive retinal image with associations with genes implicated in hair, eye and skin colour. Although race and ethnicity are believed to determine the biological phenotype of retinal pigmentation, it is likely that the RPS is a more precise measure of pigmentation. This may have implications for AI algorithm development, testing, and for inclusion and algorithmic fairness across all fields of medicine that use retinal imaging as a biomarker.

## Methods
### Ethics
We analysed data from UK Biobank participants who as part of their examinations underwent enhanced ophthalmic review. Ethics approval was obtained by the Northwest Multi-centre Research Ethics Committee (REC reference number 06/MRE08/65; approved project number 28541), our research adhered to the tenets of the Declaration of Helsinki. Informed consent was obtained from all study participants and all participants were free to withdraw from the study at any time[58].

The EPIC-Norfolk Eye Study was carried out following the principles of the Declaration of Helsinki and the Research Governance Framework for Health and Social Care and was approved by the Norfolk Local Research Ethics Committee (identifier: 05/Q0101/191) and the East Norfolk and Waveney National Health Service Research Governance Committee (identifier: 2005EC07L). All participants gave written informed consent. The study protocol is available online at https://www.epic-norfolk.org.uk/.

The Tanzanian imaging data was collected following review and approval by the Tanzanian National Institute for Medical Research (Reference id: NIMR/HQ/R.8a/Vol.IX/2402), the Kilimanjaro Christian Medical Centre (Reference number: 776), and the London School of Hygiene & Tropical Medicine Ethics Committees (Reference number: 10172).

The Australian dataset was approved by the Western Australian Aboriginal Health Ethics Committee (Reference number: 864).

The Chinese dataset was collected as part of a private dataset and ethical collection was enforced by the original dataset creators. The authors state the publishing of the dataset follows the ethical and privacy rules of China[27].

## Study population

The UK Biobank is a national research resource aiming to improve prevention, diagnosis, and treatment of a wide range of diseases. More than 500,000 people aged 37–73 were recruited at 22 study assessment centres across the UK between January 2006 and October 2010. Further details of the overall study protocol and protocols for individual tests are available online (https://biobank.ndph.ox.ac.uk/ukb/index.cgi).

All participants completed a detailed touchscreen questionnaire on demographic, clinical and lifestyle-related information. Sex was acquired from a central NHS registry at recruitment, but in some cases updated by the participant, hence this field contained a mixture of the sex the NHS had recorded for the participant and self-reported sex. Analysis of sex on RPS was performed in supplemental figures. The choices for ethnic background were categorised as white, mixed, Asian or Asian British, Black or Black British, Chinese, or other ethnic group. Participant postcode at the time of recruitment was used to determine Townsend Deprivation Index (TDI), based on the corresponding output area from the preceding national census; a higher positive score implies a greater degree of deprivation. Medical history was obtained through verbal interview with a trained nurse, including the date of first diagnosis for non-cancer and cancer illnesses, as well as any major operations. Participants gave broad consent for prospective data linkage to national electronic health records (EHR) and registries, including hospital episode statistics, death register and cancer register. Linkage to primary care records is currently available for approximately 45% of the cohort (~230,000 participants, up to 2016 or 2017 depending on data supplier). Further details of the overall study protocol and protocols for individual tests are available online (https://biobank.ndph.ox.ac.uk/ukb/index.cgi).

The EPIC-Norfolk Eye Study is a study of 8623 participants from Norfolk, England and was added onto the EPIC Cohort[29]. The EPIC study is a collaborative study involving 10 countries that began participant recruitment in 1989[59]. The EPIC-Norfolk, a United Kingdom branch of this study, comprises a population-based cohort of 25 639 participants between 40 and 79 years of age at enrolment recruited from 35 participating general practices in Norfolk, United Kingdom. Baseline examinations were carried out between 1993 and 1997[60].

The Tanzanian dataset consisted of images acquired from people attending a diabetic eye screening service in the Kilimanjaro region of northern Tanzania between June 2017 and August 2018. All participants were African and had a diagnosis of diabetes mellitus. A total of 2076 retinal photographs of 1345 eyes from 690 people with diabetes comprise the dataset. Only images that were graded as having no, or mild, retinopathy were included in the analyses.

The Australian dataset consisted of images acquired from a single Aboriginal Community Controlled Health Service located within a metropolitan area of Perth, Western Australia. Participants were Aboriginal people with diabetes mellitus attending a retinal screening service. Retinal photographs of 1682 eyes of 864 people were acquired consecutively between July 2013 and October 2020. Only images that were graded as having no retinopathy or mild retinopathy were included in this study.

The Chinese dataset is the publicly available ODIR dataset. In brief, it is a labelled collection of manually curated colour fundus photos with a wide range of disease and pathology. There are 10,000 images from 5000 individuals from 487 clinical hospitals in 26 provinces across China[27]. From this dataset, we included only the 3098 normal fundus images.

## Ophthalmic assessment

In the UK Biobank, more than 133,000 participants underwent an enhanced ophthalmic assessment between 2009 and 2010 at 6 assessment centres, including visual acuity (with the LogMAR scale), refractive error and intraocular pressure (IOP) measurements, as well as ophthalmic imaging[61]. Baseline best corrected VA was measured using a computerised semi-automated system at 3 metres distance. Autorefraction was performed using an RC5000 Auto refractor keratometer (Tomey, Nagoya, Japan). The spherical equivalent was calculated by adding the sum of the spherical power and half of the cylindrical power. Single-field undilated colour fundus photographs (45° field of view, centred to include both optic disc and macula) and macular optical coherence tomography (OCT) scans were captured using a digital Topcon-1000 integrated ophthalmic camera (Topcon 3D OCT-1000 Mark II, Topcon Corp., Tokyo, Japan). Imaging was performed after visual acuity, non-cycloplegic autorefraction, and intraocular pressure (IOP) measurement. The right eye was imaged first.

An ophthalmic examination for the EPIC-Norfolk participants was performed on 8623 of participants as part of the third health examination, carried out between 2004 and 2011. Fundus photography was acquired using a TRC-NW6S non-mydriatic retinal camera with a 10 megapixel Nikon D80 camera (Nikon corporation, Tokyo, Japan) via the IMAGEnet Telemedicine System (Topcon Corporation, Tokyo, Japan)[29].

The Tanzanian population underwent colour fundus photography using a Topcon TRC-NW8 non-mydriatic retinal camera with a 12.3 megapixel Nikon D90 camera (Nikon corporation, Tokyo, Japan). All images were acquired after dilation with 1% tropicamide and followed a standardised protocol whereby two 45° images (disc centred and macular centred) were collected per eye.

The Indigenous Australian cohort underwent colour fundus photography and OCT imaging acquired using a Topcon ophthalmic camera (Topcon 3D-OCT1 Maestro, Topcon Corp., Tokyo, Japan). Images were undilated, 45°, single-field, macular-centred photographs.

The Chinese dataset is acquired with multiple different cameras with varying fields-of-view. The information about camera type and field-of-view was not publically available in the data labels.

## Retinal pigment score

Each fundus image was run through the AutoMorph pipeline[23] to create a segmentation mask of the retinal vasculature and optic disc. The deep learning models used for each step of the segmentation pipeline are described in detail in their manuscript. In summary, each image is pre-processed and then passed through an image quality classifier that was pre-trained on the EyePACs dataset. All images of insufficient quality are excluded from the subsequent segmentation steps. The binary vessel segmentation module is an ensembled UNet architecture and the optic disc segmentation is a cascaded UNet with a ResNet backbone. We modified the AutoMorph code by changing the file system organisation between input and output nodes, adding in error handling, and reducing output file size.

We took the segmentation masks for the disc segmentation and the binary vessel segmentations from the AutoMorph modules and added them together to make a combined disc and vessel mask. The background of the fundus image was identified by finding all pixels that were at or below the 0.5 percentile of the distribution of all grayscale pixels from the input image. The background mask was added to the combined vessel and disc mask. This mask was then successively dilated using a 2-dimensional binary structuring kernel with connectivity of two. The number of dilation iterations was four multiplied by the pixel width of the image divided by 600 rounded to the nearest integer, which was derived empirically. All pixels not contained in dilated masks of the background/disc/segmentation mask were used to create a new retinal background mask.

From the retinal background mask, we found the median RGB pixel value and converted it into the CIELAB colour space[24] which is a colour space designed to have luminance (L vector) stored in a separate vector from chromaticity (a,b vectors). To reduce the effect of

illumination on the measured colour from the retina, we only used the a,b coordinates from the CIELAB space and ignored the L vector. To transform the two-dimensional a,b chromaticity vectors for each eye, we used a principal component analysis model to perform dimensionality reduction. For each dataset, a two-component PCA model was fitted to the median a,b value of the retinal background for all images in the dataset. Then, each eye's median a,b value was transformed with the PCA model along the eigenvector with the greatest eigenvalue. This new transformed vector was stored as the 1-dimensional RPS vector. Figure 1 represents a schematic of the pipeline. Sensitivity analyses were conducted to test for RPS performance across different background colour image pigmentation by calculating the vessel and optic disc mask area from the total pixel image area by ethnicity and by RPS quintiles. The effect of increasing or decreasing the vessel and disc segmentation area on the RPS was examined with a random selection of either a binary erosion or binary dilation for 1 iteration with a 3×3 kernel to the combined vessel and disc masks for a random selection of 100 fundus images from each of the white, Black, Asian, Chinese and Mixed ethnic groups of the UK Biobank.

The image analysis to derive the Retinal Pigment Score (RPS) was performed with Python, version 3.8[62] and PyTorch, version 1.7.0[63]. The code to derive RPS is publicly available at: https://github.com/uw-biomedical-ml/retinal-pigmentation-score.

RPS reliability was assessed in an independent dataset of 27 eyes from 24 patients (22 healthy patients and 5 patients with mild nonproliferative diabetic retinopathy). The images were captured on Topcon 3D OCT-1 Maestro. One-way intraclass correlation (ICC) was calculated between 2 repeat images from the same eye.

We assessed the mean difference, standard deviation and ICC between the RPS of right and left eyes from the same patients within the UK Biobank dataset. Additionally, we assessed the mean, standard deviation and ICC of the RPS when the L, a and b vectors were used to fit the principal component analysis (PCA).

## Genome-wide association study

Genome-wide association study (GWAS) was performed to assess potential genetic associations with standardised mean RPS (average score between right and left eyes per participant). A beta coefficient of 1 therefore corresponds to a 1 standard deviation increase in standardised mean RPS. Analyses were conducted using a generalised linear mixed model, adjusting for age, sex and the first ten principal components. The initial discovery GWAS analysis was performed in the UK Biobank cohort. Lead variants reaching genome-wide significance ($p < 5 \times 10^{-8}$) were re-evaluated in a replication GWAS analysis, conducted in the EPIC-Norfolk cohort. A Bonferroni-adjusted replication significance threshold was set at $p = 0.05/17$.

Lead variants were furthermore investigated for previously identified associations with hair, skin and eye colour, by manually referring to the Open Targets Genetics[64,65] and PhenoScanner[66,67] databases and the results are listed in Table 1.

Full details for genotyping and imputation in the UK Biobank cohort have been described previously[68]. In brief, genotype calling was performed using two arrays: the UK BiLEVE Axiom array (~50,000 participants) and the UK Biobank Axiom array (~450,000 participants). Marker positions are in GRCh37 coordinates. There were 805,426 markers available in the released data after quality control. Genotype imputation was then performed using a combined Haplotype Reference Consortium and UK10K reference panel, expanding the number of testable variants to ~96 million. The majority of the EPIC-Norfolk cohort were also genotyped using the Affymetrix UK Biobank Axiom Array, however, genotyping for a small subset was undertaken using the Affymetrix GeneChip Human Mapping 500 K Array Set.

The following exclusions were applied for sample quality control: individuals with relatedness corresponding to third-degree relatives or closer, excess of missing genotypes or more heterozygosity than expected. The GWAS analysis was furthermore restricted to individuals of European ethnicity only. The following exclusions were applied for variant-level quality control: call rate <95%, Hardy–Weinberg equilibrium $p < 1 \times 10^{-6}$, posterior call probability <0.9, INFO score <0.9 and minor allele frequency <0.01.

The SNP2GENE function within FUMA[69] was used to perform positional and expression quantitative trait locus (eQTL) gene mapping/prioritisation. SNPs in high LD ($r^2 > 0.6$) with any independent lead variant were positionally mapped to genes located within 10 kb. Variants were also mapped to a set of prioritised genes within 1 Mb if associated with the expression of those genes in retina (reported in EyeGEx), skin (reported in TwinsUK, GTEx v8) and dermal fibroblasts (reported in GTEx v8). To test the prioritised genes for enrichment in biological pathways, FUMA's GENE2FUNC function was applied, using hypergeometric mean pathway analysis against gene sets obtained from MsigDB and WikiPathways. Multiple testing correction (Benjamini-Hochberg method) per data source of tested gene sets was performed and gene sets with adjusted $P \leq 0.05$ and >1 overlapping genes were reported by default.

## Phenome-wide association study

A phenome-wide association (PheWAS) analysis was conducted within the discovery UK Biobank GWAS sub cohort using 308 CALIBER codelists drawing on the following diagnostic records: verbal interview responses, linked hospital episode statistics, death register and primary care records. Read 2, ICD-10 and OPCS-4 clinical codelists were minimally adapted from the CALIBER Portal[70]. The former two coding systems were expanded to Read 3 and ICD-9 equivalents, respectively, using the mapping files provided by UK Biobank Resource 592.

The PheWAS analysis performed logistic regression to assess potential disease associations with standardised mean RPS (average score between right and left eyes per participant), adjusting for age at baseline assessment and self-reported sex. All available diagnostic records, both before and after the date of attendance for retinal imaging, were included. Conditions with fewer than 200 cases were excluded. Associations meeting the Bonferroni-corrected $p$-value threshold ($p = 0.05/308$) were considered phenome-wide significant.

## Mendelian randomisation

Mendelian randomisation (MR) is a technique for evaluating causality between exposure and outcome variables by utilising genetic variants as instrumental variables (IV)[71]. In comparison to traditional epidemiological methods, MR is relatively immune to bias from confounding and reverse causation. The random allocation of genetic variants at conception is analogous to the random allocation of an intervention in a randomised controlled trial. For the results from an MR analysis to be valid, the IV must satisfy three critical assumptions: (1) the IV must be associated with the exposure; (2) the IV must not be associated with confounders of the exposure-outcome association; (3) the IV must only affect the outcome via the exposure and not through alternative pathways. Provided these criteria are met, the estimates from a MR analysis reflect the causal association between a genetically determined risk factor (here, a genetic predisposition to increased retinal pigmentation) and the development of a particular outcome over the course of a lifetime.

Two-sample MR analyses were conducted drawing on the discovery RPS GWAS results (exposure) and summary statistics from the independent FinnGen cohort[72] (data release 8) for selected outcomes of interest, as highlighted by the RPS PheWAS analysis. The IV for retinal pigmentation was constructed by selecting all conditionally independent genome-wide significant single nucleotide

polymorphisms (SNPs) from the discovery RPS GWAS analysis, and performing clumping to exclude those with linkage disequilibrium $R^2 > 0.01$ and within 10,000 kb, using the 1000 Genomes Project European reference population[73]. Palindromic SNPs with minor allele frequency >0.46 were excluded. Effect alleles were harmonised across exposure and outcome datasets.

The main MR analyses were performed using a multiplicative random-effects inverse-variance weighted (IVW) approach[74]. This method provides precise and efficient estimates but is sensitive to invalid IVs and pleiotropy[75]. We therefore conducted sensitivity analyses using three alternative MR methods: weighted median[76], weighted mode[77] and MR-Egger[78]. Each method makes different assumptions about the nature of pleiotropy and consistent estimates across methods strengthens causal inferences[79].

Under the IVW method, we calculated the mean F-statistic as an indicator of instrument strength (a value > 10 is usually considered a strong instrument)[80]. We assessed for heterogeneity with the $I^2$ and Cochran's $Q$ statistics in the IVW model and with Rucker's $Q'$ statistic in MR-Egger regression. The $I^2_{GX}$ statistic is an indicator of expected relative bias (or dilution) of the MR-Egger causal estimate[81]. In MR-Egger regression, a significant difference of the intercept from zero is evidence for average directional horizontal pleiotropy[78].

## Statistical analysis

All measurements were repeated from the same sample unless otherwise reported. Linear regression models with standardised RPS (z-score) adjusting for age, sex, self-reported ethnicity (categorised as white, Black, Asian, mixed, Chinese, or other), hair colour (categorised as blonde, red, light brown, dark brown, black and other), skin colour (categorised as very fair, fair, light olive, dark olive, brown and black), spherical equivalent, height, TID (scores categorised in quintiles where a higher quintile implies a greater degree of deprivation), and UK Biobank assessment centre were used to examine associations with RPS. Collinearity was examined using variance inflation factor testing on the final model with adjusted generalised standard error inflation factors[82]. Missing data points were categorised as "Missing" within each variable. Formal two-way tests for interaction were examined for ethnicity and height, and ethnicity and age. Sensitivity analyses were conducted with three stratified linear regression models for the three main ethnic groups (white, Black, and Asian ethnic groups).

The GWAS analysis was performed using REGENIE software[83]. GCTA-COJO was used to prioritise lead variants[84]. Genomic inflation factor and heritability estimates were calculated using the LDSC tool[85] and pre-calculated LD scores for European ancestry (https://data.broadinstitute.org/alkesgroup/LDSCORE/eur_w_ld_chr.tar.bz2). All other statistical analyses and were performed in R (R for GNU macOS, Version 4.2.0, The R Foundation for Statistical Computing, Vienna, Austria)[86]. R packages used included PheWAS[87], TwoSampleMR[88], MendelianRandomization[89], ukbwranglr[90], codemapper[91], targets[92], tarchetypes[93], tidyverse[94], workflowr[95], flextable[96], gtsummary[97] and knitr[98].

## Inclusion and ethics statement

All collaborators of this study have fulfilled the criteria for authorship required by Nature Portfolio journals have been included as authors. Roles and responsibilities were agreed among collaborators ahead of the research. This research was not severely restricted or prohibited in the setting of the researchers, and does not result in stigmatisation, incrimination, discrimination or personal risk to participants. Local and regional research relevant to our study was considered in citations.

## Reporting summary

Further information on research design is available in the Nature Portfolio Reporting Summary linked to this article.

## Data availability

Access to the UK Biobank is restricted to safeguard the privacy of the participants and requires an application. The restrictions depend on the level of access granted. One can apply for access on their website. You cannot share the UK Biobank data with researchers who are not registered with the UK Biobank. Registrations are reviewed within 10 working days of submission. The length of access depends on the access granted. Access to the EPIC-Norfolk Eye study is restricted and requires an application because of the desire to safeguard the privacy of participants. One can request access via the EPIC-Norfolk Management Committee. The data is available to researchers with relevant scientific and ethics approvals for their research, including those in other countries and in commercial companies who are looking for new treatments or laboratory tests. Applications are generally reviewed within 1 month. The Tanzanian fundus photo dataset was transferred to LSHTM under a formal data transfer agreement with the National Institute for Medical Research in Tanzania. This agreement stipulates that the dataset be used for teaching or academic research purposes only. Requests for access to the dataset can be made to Charles Cleland (charles.cleland@lshtm.ac.uk) and replies will be within ten working days. If access to the dataset is granted it will be for a period of six weeks. The Australian dataset is available under restricted access in order to safeguard participant privacy. This dataset, also known as the Derbarl Yerrigan Health Service data, is a First Nations of Western Australia diabetic screening dataset. This dataset is subject to ethical approval for use by the Ethics Committee operated by the Aboriginal Health Council of Western Australia (AHCWA). A written request will be considered and responses should be returned in less than one month. Access to the data is subject to further ethics applications to AHCWA and the duration of access will depend on the applications. Please reach out to angus.turner@uwa.edu.au for inquiries. The Chinese dataset is a subset of the publically available ODIR dataset and can be accessed as described in the corresponding manuscript. (Li et al. 2021). The raw data used in this study are protected and are not available due to data privacy laws. The data generated for figures presented in this study are provided in the Source Data file. Retinal pigment scores for UK Biobank participants will be made available to approved UK Biobank researchers as a returned dataset (https://biobank.ndph.ox.ac.uk/ukb/docs.cgi?id=1). FinnGen genome-wide association study (GWAS) summary statistics are publicly available online (https://www.finngen.fi/en/access_results). Summary statistics from the GWAS analyses presented in this study will be made publicly available from the NHGRI-EBI Catalog of human genome-wide association studies (https://www.ebi.ac.uk/gwas/). Source data are provided with this paper.

## Code availability

The code to derive RPS is publicly available at https://github.com/uw-biomedical-ml/retinal-pigmentation-score.

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

## Acknowledgements

The authors would like to thank the 500,000 UK Biobank participants, the participants and the investigators of the FinnGen study, the EPIC-Norfolk study, the ODIR dataset, the Australian dataset and the Tanzanian dataset; and the funders of this project. We thank Topcon Healthcare, particularly Mary Durbin, Ted Spaide, and Mark Evans, UK & Ireland Service and Operations Manager, and Leigh Maher, IT and Systems, for their invaluable contribution in providing an independent dataset for standardisation techniques. The EPIC-Norfolk study (DOI 10.22025/2019.10.105.00004) has received funding from the Medical Research Council (MR/N003284/1 and MC-UU_12015/1) and Cancer Research UK (C864/A14136). The genetics work in the EPIC-Norfolk study was funded by the Medical Research Council ((MC_PC_13048)). We are grateful to all the participants who have been part of the project and to the many members of the study teams at the University of Cambridge who have enabled this research. The authors would like to acknowledge Professor Nick Wareham for his assistance in access to the EPIC-Norfolk Eye study. C.E. and A.T. received a proportion of their financial support from the UK Department of Health through an award made by the National Institute for Health Research to Moorfields Eye Hospital NHS Foundation Trust and UCL Institute of Ophthalmology for a Biomedical Research Centre for Ophthalmology. C.E. receives a proportion of financial support to Moorfields Eye Hospital NHS Foundation Trust from the Lowy Medical Research Institute. C.S.L. receives grant funding from NIH/NIA R01AG060942, U19AG066567, OT2OD032644, and the Klorfine Family Endowed Chair, and the Karalis Johnson Retina Center. A.Y.L. has received an unrestricted and career development award from Research to Prevent Blindness, Latham Vision Science Awards, NEI/NIH K23EY029246, OT2OD032644, the C. Dan and Irene Hunter Endowed Professorship, and the Karalis Johnson Retina Center. A.O.-B. is supported by the Lowy Medical Research Institute, La Jolla, California. A.P.K. is supported by a UK Research and Innovation Future Leaders Fellowship, an Alcon Research Institute Young Investigator Award and a Lister Institute for Preventive Medicine Award. This research was supported by the NIHR Biomedical Research Centre at Moorfields Eye Hospital and the UCL Institute of Ophthalmology. A.N.W. is supported by the Wellcome Trust (220558/Z/20/Z; 224390/Z/21/Z). P.A.K. is supported by a UK Research & Innovation Future Leaders Fellowship (MR/T019050/1) and The Rubin Foundation Charitable Trust. M.J.B. is supported by a Wellcome Trust Senior Research Fellowship (207472/Z/17/Z), and a Wellcome Trust Collaborative Award (206275/Z/17/Z). K.V.S. is supported by a UCL Overseas Research Scholarship and grants from Fight for Sight (1956A) and the Desmond Foundation. The research was supported by the National Institute for Health and Care Research (NIHR) Biomedical Research Centre based at Moorfields Eye Hospital NHS Foundation Trust, UCL Institute of Ophthalmology, and the NIHR Birmingham Biomedical Research Centre. The views expressed are those of the author(s) and not necessarily those of the NHS, the NIHR or the Department of Health and Social Care.

## Author contributions

A.E.R. wrote the code for generating the RPS. A.O.B. performed statistical tests of clinical variables. A.N.W. performed the initial GWAS on the UKBiobank cohort. A.N.W., K.V.S., M.I.B., A.P.K., R.L., P.J.F. performed additional GWAS analysis on the UKBiobank cohort and the EPIC-Norfolk cohort. C.R.C., W.U.M., M.J.B., A.B. provided, evaluated, and maintained the Tanzanian dataset. P.A.K., M.A.C., A.W.T. provided, evaluated and maintained the Australian dataset. A.E.R., A.O.B., C.S.L. A.Y.L., A.T., C.S.L., Y.W. wrote the manuscript along with input from all other coauthors.

## Competing interests

A.P.K. has acted as a paid consultant or lecturer to Abbvie, Aerie, Allergan, Google Health, Heidelberg Engineering, Novartis, Reichert, Santen,Thea and Topcon. A.Y.L. reports support from the US Food and Drug Administration, grants from Santen, Carl Zeiss Meditec, and Novartis, personal fees from Genentech, Topcon, and Verana Health, outside of the submitted work; This article does not reflect the opinions of the Food and Drug Administration. A.T. report grants from Bayer and Novartis and personal fees from Abbvie, Allegro, Annexon, Apellis, Bayer, Heidelberg Engineering, Iveric Bio, Kanghong, Novartis, Oxurion, Roche/Genentech, Thea. C.E. reports personal fees from Heidelberg Engineering, Boehringer Ingelheim, and Inozyme pharmaceuticals outside of the submitted work. P.A.K. has acted as a consultant for Retina Consultants of America, Topcon, Roche, Boehringer-Ingleheim, and Bitfount and is an equity owner in Big Picture Medical. He has received speaker fees from Zeiss, Novartis, Gyroscope, Boehringer-Ingleheim, Apellis, Roche, Abbvie, Topcon, and Hakim Group. He has received travel support from Bayer, Topcon, and Roche. He has attended advisory boards for Topcon, Bayer, Boehringer-Ingleheim, RetinAI, and Novartis. P.J.F. has acted as a consultant for Alphasights, GLG, Google Health, Guidepoint, PwC, Santen. A.B. is Founder and CEO of not-for-profit Peek Vision and receives a salary. The remaining authors declare no competing interests.

## Additional information

# UK Biobank Eye and Vision Consortium

Naomi Allen[12], Tariq Aslam[13], Denize Atan[14], Konstantinos Balaskas[3], Sarah Barman[15], Jenny Barrett[16], Paul Bishop[17], Graeme Black[17], Tasanee Braithwaite[5], Roxana Carare[12], Usha Chakravarthy[18], Michelle Chan[3], Sharon Chua[3], Alexander Day[3], Parul Desai[3], Baljean Dhillon[19], Andrew Dick[14], Alexander Doney[20], Sarah Ennis[21], John Gallacher[22], David Ted Garway-Heath[3], Jane Gibson[12], Jeremy Guggenheim[23], Chris Hammond[24], Alison Hardcastle[3], Simon Harding[25], Ruth Hogg[18], Pirro Hysi[24], Gerassimos Lascaratos[26], Thomas Littlejohns[12], Andrew Lotery[27], Phil Luthert[3], Tom MacGillivray[19], Sarah Mackie[16], Savita Madhusudhan[25], Bernadette McGuinness[18], Gareth McKay[18], Martin McKibbin[16], Tony Moore[3], James Morgan[23], Eoin O'Sullivan[26], Richard Oram[28], Chris Owen[29], Praveen Patel[3], Euan Paterson[18], Tunde Peto[18], Axel Petzold[3], Nikolas Pontikos[3], Jugnoo Rahi[30], Alicja Rudnicka[29], Naveed Sattar[31], Jay Self[27], Panagiotis Sergouniotis[17], Sobha Sivaprasad[3], David Steel[32], Irene Stratton[33], Nicholas Strouthidis[3], Cathie Sudlow[34], Zihan Sun[3], Robyn Tapp[35], Dhanes Thomas[3], Emanuele Trucco[20], Ananth Viswanathan[3], Veronique Vitart[34], Mike Weedon[28], Katie Williams[24], Cathy Williams[14], Jayne Woodside[18], Max Yates[36] & Yalin Zheng[25]

[12]Nuffield Department of Population Health, University of Oxford, Oxford, UK. [13]Manchester Royal Eye Hospital, The University of Manchester, Manchester, UK. [14]Bristol Eye Hospital, University of Bristol, Bristol, UK. [15]Department of Computer Science and Mathematics, Kingston University, London, UK. [16]School of Medicine, University of Leeds, Leeds, UK. [17]Division of Evolution, Infection and Genomics, The University of Manchester, Manchester, UK. [18]Centre for Public Health, School of Medicine, Dentistry and Biomedical Sciences, Queen's University Belfast, Belfast, UK. [19]Centre for Clinical Brain Sciences, University of Edinburgh, Edinburgh, Scotland. [20]Pat Macpherson Centre for Pharmacogenomics and Pharmacogenetics, Division of Population Health & Genomics, School of Medicine, University of Dundee, Dundee, UK. [21]Human Development and Health, Faculty of Medicine, University Hospital Southampton, Southampton, Hampshire, UK. [22]Dementias Platform UK, Oxford, UK. [23]School of Optometry & Vision Sciences, Cardiff University, Cardiff, UK. [24]Department of Twin Research and Genetic Epidemiology, King's College London, London, UK. [25]Department of Eye and Vision Science, Institute of Life Course and Medical Sciences, University of Liverpool, Liverpool, UK. [26]Department of Ophthalmology, King's College Hospital NHS Foundation Trust, London, UK. [27]Clinical and Experimental Sciences, Faculty of Medicine, University of Southampton, Southampton, UK. [28]University of Exeter College of Medicine & Health, Exeter, UK. [29]Population Health Research Institute, St George's, University of London, London, UK. [30]Population, Policy and Practice Research and Teaching Department, Great Ormond Street Institute of Child Health, University College London, London, UK. [31]School of Cardiovascular and Metabolic Health, University of Glasgow, Glasgow, UK. [32]Biosciences Institute, Faculty of Medical Sciences, Newcastle University, Newcastle upon Tyne, UK. [33]Gloucestershire Retinal Research Group, Cheltenham General Hospital, Gloucestershire Hospitals NHS Foundation Trust, Cheltenham, UK. [34]Centre for Medical Informatics, Usher Institute of Population Health Sciences and Informatics, University of Edinburgh, Edinburgh, UK. [35]Research Centre for Intelligent Health Care, Coventry University, Coventry, UK. [36]Norwich Epidemiology Centre, Norwich Medical School, University of East Anglia, Norwich, UK.

