## [Peer Review File · Nature Communications]

Reviewers' Comments:

Reviewer #1:

Remarks to the Author:

Interesting work on a really important topic. Although the motivation and approach are clear, there are a few questions/concerns about the methodology and conclusions that could be better clarified.

-- Populations studies- although the UKBB does contain people of east Asian (Chinese) origin, a more representative dataset of that population would be good to validate the generalizability of a single scale to capture pigmentation.

-- Robustness studies are largely missing. The impacts of illumination, field of view, image quality etc. are unclear. How robust is the RPS to acquisition differences?

-- Repeatability and reproducibility studies are key to ensuring the validity of RPS. Ideally, the same person would be imaged multiple times under different conditions (illumination, FoV etc.) and bounds of the test-retest agreement would be assessed.

-- Fundamentally, it is unclear if converting color (with values across different hues) to a single scale is appropriate. Retinal pigmentation, even more than skin perhaps can have different balances between red and yellow for instance. These could be lost in converting to a single scale. It would be interesting to better understand how different hues and shades of red/yellow manifest in the dimensionality reduced CIELAB space. This would especially be interesting in comparing East Asian with South Asian populations where the RGB balances are different.

-- Along those lines, when PCA was performed what fraction was explained by the single dimension? i.e. what are the Eigen values of the first 2 dimensions? Is there convincing evidence that a single dimension suffices? The near identical distribution of RPS scores for Black and Chinese seems questionable (probably a consequence of use just 1 dimension?)

-- The association between RPS and clinical variables is somewhat surprising. What plausible biological explanation might be considered to explain RPS and height and age? Could this be potentially confounding? Are the relative age and height distributions similar in all populations (presumably not?). Were interactions considered in the model? Even though these were adjusted for, it would be helpful to look at the plots of RPS and age or height in different populations.

-- What was the average difference between RPS of the two eyes (paired difference)?

-- Could you comment on the implications of the replication study (only 2 of 17 achieved genome wide significance)?

Overall, a potentially useful approach but the risk of overfitting, confounding and loss of too much information are concerns. Robustness studies to evaluate and better understand the finishing are important.

Reviewer #2:

Remarks to the Author:

This manuscript derives a new metric called retinal pigment score (RPS) that quantifies the degree of pigmentation from fundus images. It is well known that pigmentation confounds processing of fundus images, i.e., detection of vessels. Similar ideas

as in this paper have previously been used to enhance fundus images (e.g., Retinal image enhancement based

on color dominance of image, Scientific Reports, 2023, and refs within). However, the construction of a image

derived score quantifying the pigmentation seems to be novel.

1. I was a bit disappointed with the results, RPS did not yield much biological insight or at least it wasn't highlighted in the paper. The main application that was highlighted was to quantify the diversity of AI training sets. Can you please elaborate, especially with respect to the fact that RPS is dataset dependent. Please also elaborate on how exactly this can be resolved with standardisation of the metric between camera types, using device-specific raw RGB values and why that wasn't done in this work.

2. In Figure 2 b the UKB scores and the scores for the Tanzanian and Australian datasets were plotted on the same plot.

Is that valid since RPS is dataset dependent? Is the difference between "Black" and "Tanzanian" real?

3. The method is clearly yielding scores that are correlated with pigmentation. However, it is easy to think of other methods that does the same. Can you give some insight into why the proposed method is reasonable, e.g., why PCA instead of finding some other direction in the a-b space?

4. Sometimes the manuscript lacks explanations and details, e.g., what are the number inside the parenthesis in line 106? Give more detail about the details of the QC for RPS. Were the RPS scores inverse normal transformed for GWAS? Why weren't the similar corrections used in GWAS, Phewas and statistical analysis?

Reviewer #3:

Remarks to the Author:

A well-researched and impactful work. The main objective of the paper is to develop the Retinal Pigmentation Score (RPS), a continuous measurement of retinal pigmentation derived from retinal images. The motivation behind the work is well-defined. The results justify the objective. It will help ensure the explainability and fairness of AI models.

One concern is that in the AutoMorph pipeline, what is the justification behind using ensembled Unet architecture and not other architectures like “Hard Attention Net (HANet)” by D. Wang et. al., “Scale-space approximated convolutional neural networks (SSANet)” by K. J. Noh et. al. for Retinal Vessel Segmentation.

The authors should present the effect of including the L vector on RPS.

Manuscript ID: NCOMMS-23-28885A-Z

Ethnicity is not biology: retinal pigment score to evaluate biological variability from ophthalmic imaging using machine learning

The authors would like to thank the editor and the reviewers for the opportunity to revise this manuscript and for their constructive feedback. We have copied the reviews on the left column, our response in the middle column, and the changes made in the manuscript on the right column. Additions to the text are displayed in text with a different colour and deletions are shown in strikethrough.

Reviewer comments	Author response	Manuscript changes
Reviewer #1		
Interesting work on a really important topic. Although the motivation and approach are clear, there are a few questions/concerns about the methodology and conclusions that could be better clarified.	We thank the reviewer for their thoughtful review and positive remarks. We have addressed the comments point by point.	
-- Populations studies- although the UKBB does contain people of east Asian (Chinese) origin, a more representative dataset of that population would be good to validate the generalizability of a single scale to capture pigmentation.	We thank the reviewer for this thoughtful comment. We agree that this is a potential limitation of the current study and, thus in response, we have identified additional datasets for analysis. The additional experiments are relevant to this comment and to a later comment related to UKBiobank self-described “Black ethnicity” according to that study protocol and participants in the additional datasets from Tanzania and Australia. We ran the RPS code on an open-source dataset called	Results Page 4 Figure 2a and b have been changed to show results from the additional 3 datasets. We have also updated the methods to describe the dataset in more detail.

	ODIR. ODIR has 5,000 fundus photographs from different cameras from different hospitals/medical centers in China. (doi: https://doi.org/10.48550/arXiv.2102.07978) This dataset does not provide detailed camera information and the authors have not responded to a request for the information at the time of submitting this response. In lieu of that information, we elected to use the UKBiobank dataset as the reference dataset to fit the RPS for the ODIR dataset for a qualitative comparison as the reviewer requested. From the qualitative comparison, we observe that the ODIR dataset falls in a similar range to that of the Chinese participants from the UKBB. We are reassured that the distribution RPS of ODIR and the Chinese UKB participants are in a similar range along the RPS scale. We have added an additional row to Figure 2a to show representative images from the Chinese (ODIR) dataset and included the distribution of RPS for ODIR into Figure 2b.	
-- Robustness studies are largely missing. The impacts of illumination, field of view, image quality etc. are unclear. How robust is the RPS to acquisition differences?	We appreciate this helpful comment. We have addressed this in three parts: image quality, illumination and field of view.	Results Page 4 Briefly, the algorithm uses published open-access deep learning models²³ to identify and exclude ungradable

Image quality is the initial step in the process and may not have been sufficiently clear to the reader. The first step of the code checks the image quality for flash artefacts, opacities and other pathology that affects the quality of the image. These images are removed from the Automorph pipeline and are not used to create the RPS (See Methods - Retinal Pigment Score, page 12, first paragraph). For clarity, we have now added this detail earlier in the results section.

Illumination plays a role in the RGB values of each pixel. To minimise the effect of illumination on the chromaticity, we used the CIE-LAB colorspace. This colorspace has illumination separated in its own vector L which is designed to be independent from the a and b vectors which represent chromaticity. We showed that adding the L vector in addition to the chromaticity vectors of a and b to fit the principal component analysis model for the RPS did not reduce the variance. Among participants in the UKBiobank that had fundus photos of both the right and left eyes, the mean and standard deviation of the intra-participant difference between right and left eye was smaller for the RPS generated with

images, create a mask of the tissue that excludes the retinal vasculature and optic nerve, then finds the average chromaticity of this background tissue.

Modified Figure : Figure 2b.

Modified Figure 2 legend:

The RGB colour of the pixel value that is converted into RPS as well as the RPS is shown at the bottom of each fundus photograph. Black spaces represent when there are no suitable images within the respective ethnicity subgroup and quintile b. Normalised kernel density estimation plot of the distribution of RPS for all participants grouped by self-reported ethnicity as reported in the UK Biobank as well as the Tanzanian, Indigenous Australian, and Chinese(ODIR) datasets. Relative frequencies are normalised so the area under each curve is equal for each ethnicity subgroup. The subpanel consists of examples where for a given RPS and the a, b values in the CIELAB colour space are constant but the L vector changes. The x-axis is shared in both subpanels.

just the a, b vectors than the RPS generated from the L, a, b vectors.

To show readers how much the L vector affects the RGB pixel values, while keeping the RPS the same we modified Figure 2 to demonstrate this phenomenon. In this figure you can see that a wide range of RGB values all condense to the same RPS, because these RGB values have the same a, b values and only varying degrees of illumination represented in the L vector.

This is reassuring that we are eliminating some of the effect of illumination on RPS.

Field of view:

All of the image datasets used in this study used standard fundus cameras, typically with a field of view of 30 to 50 degrees visual angle (the images from the ODIR dataset were qualitatively similar in this respect, which can be easily determined by the retinal anatomic landmarks visible in the image). Wider angle cameras are available, but are not available as large scale, open source datasets and are not typically used for e.g. population screening for diabetic retinopathy but used in small specialist eye clinics. The purpose of the RPS is to generate a meaningful metric to assess population based

	screening tools, therefore we elected not to pursue wide angle fundus photos for our study. The field of view is determined by the camera type, which is described in our Methods section (Methods, Ophthalmic assessment, page 11).	
-- Repeatability and reproducibility studies are key to ensuring the validity of RPS. Ideally, the same person would be imaged multiple times under different conditions (illumination, FoV etc.) and bounds of the test-retest agreement would be assessed.	We agree that this is important. We were able to assess the reliability of test-retest from an additional dataset consisting of 27 eyes from 24 patients of which 22 were normal and 5 had a diagnosis of mild nonproliferative diabetic retinopathy. We elected to use intraclass correlation (ICC) as a way to assess the relationship between the measurements of the right and left eye because we assume that the RPS should be similar between laterality. A high ICC would suggest adequate reproducibility and reliability. The one-way consistency intraclass correlation coefficient was 0.92 with 95% confidence interval [0.834, 0.963] and p-value <0.001. This suggests that there is a high degree of reliability in the RPS for measuring the same eye. We have incorporated the methodology for analysis and results in the corresponding Methods and Results section as seen in our change column.	Results Retinal Pigment Score reliability Page 4 We assessed RPS reliability in an independent dataset of 27 eyes from 24 patients with two images per eye). The one-way consistency intraclass correlation coefficient (ICC) was 0.92 (95% CI 0.834, 0.963, p-value <0.001). Among 30,407 participants in the UK Biobank that had available imaging, the mean and standard deviation of the difference in RPS between the right and left eye was -1.36 (8.30) and the one-way ICC was 0.788 (95% CI: 0.784, 0.792).. In contrast, the use of the L, a, b vectors to calculate RPS among the same group of participants yielded a mean score of -2.40 (10.70) and a one-way ICC of 0.757 (95% CI: 0.753, 0.762). Methods Page 12 RPS reliability was assessed in an independent dataset of 27 eyes from 24 patients (22

		healthy patients and 5 patients with mild non-proliferative diabetic retinopathy). The images were captured on Topcon 3D OCT-1 Maestro. One-way intraclass correlation (ICC) was calculated between 2 repeat images from the same eye. We assessed the mean difference, standard deviation and ICC between the RPS of right and left eyes from the same patients within the UK Biobank dataset. Additionally we assessed the mean and standard deviation of the RPS when the L,a and b vectors were used to fit the PCA.
-- Fundamentally, it is unclear if converting color (with values across different hues) to a single scale is appropriate. Retinal pigmentation, even more than skin perhaps can have different balances between red and yellow for instance. These could be lost in converting to a single scale. It would be interesting to better understand how different hues and shades of red/yellow manifest in the dimensionality reduced CIELAB space. This would especially be interesting in comparing East Asian with South Asian populations where the RGB balances are different.	With the RPS, we wanted a score with a single dimension to aid in further downstreams tasks such as evaluating for algorithmic bias while still preserving the variance of the colour space data. The proportional eigenvalues (specific eigenvalue / sum of all eigenvalues) of the first two dimensions were 0.899 and 0.101. We transformed the a,b data into the RPS using the first dimension of the PCA. This suggests that most of the variance is explained by the first dimension of the PCA. This reassures us that a single dimension suffices to represent retinal pigment.	Results Retinal Pigment Score Page 4 The median age (IQR) was 56 years (49-63) and 92% (40,704/44,320) of participants self-described their ethnicity as white. The proportional eigenvalues of the first two dimensions were 0.899 and 0.101. The median RPS was -0.82 (-9.89, 10.39).

-- Along those lines, when PCA was performed what fraction was explained by the single dimension? i.e. what are the Eigen values of the first 2 dimensions? Is there convincing evidence that a single dimension suffices? The near identical distribution of RPS scores for Black and Chinese seems questionable (probably a consequence of use just 1 dimension?)	The overlap of the RPS distributions with any of the self-reported ethnicities, including those describing themselves as Black or Chinese supports one of the main points of this manuscript. We refer the reviewer to Figure 2a. Observing the retinal photographs, it is relatively easy to subjectively ascribe a photograph to the RPS quintiles (“less pigmented” vs “more pigmented”) represented by the columns of images. It is a much more difficult task to ascribe the ethnicity label to the rows of images. This is a graphical representation of the underpinning concept of this manuscript, which is that the socio-political construct of ethnicity does not determine a biological phenotype - retinal pigmentation. Although it is reported that patients with lighter skin colour have more reddish-orange funduses [https://journals.lww.com/international-ophthalmology/Fulltext/2003/43040/Racial_and_Ethnic_Differences_in_Ocular_Anatomy.4.aspx, https://iovs.arvojournals.org/article.aspx?articleid=2177379], to our knowledge there have not been any studies that quantitatively compare the fundus pigmentation in fundus
---	---

	photographs of Black and Chinese people. Further evidence of the overlap in pigment phenotypes is presented in our figure 2 (quintiles of RPS by ethnicity). All ethnicity groups in the 4 datasets presented in our paper show a wide distribution of RPS, so that, other than at the extremes, RPS values would not differentiate ethnic groups. Previous studies of difference based on examination of ocular tissue or human labels of images have utilised very small sample sizes compared with our study.	
-- The association between RPS and clinical variables is somewhat surprising. What plausible biological explanation might be considered to explain RPS and height and age? Could this be potentially confounding? Are the relative age and height distributions similar in all populations (presumably not?). Were interactions considered in the model? Even though these were adjusted for, it would be helpful to look at the plots of RPS and age or height in different populations.	We have included the adjusted mean of RPS for the three main ethnic groups in the dataset for height and age deciles. Height is positively associated with refractive status and axial length(https://doi.org/10.1371/journal.pone.0043172). Larger eyes in taller people can be an underpinning factor for this association, and although adjusting for axial length would have been ideal, these measurements are unfortunately not available in the UK biobank. There were no significant interactions between height and ethnicity. We cannot rule out the presence of residual confounding when assessing the association with age and	Results Association of RPS with Clinical Variables Page 5 Every 5-year rise in age was associated with a 0.02 increase in RPS ($p = 1.3 \times 10^{-8}$), and every 5 cm increase in height conferred a -0.02 change in RPS ($p = 3.6 \times 10^{-8}$). Supplementary figure 2 shows mean RPS adjusted for sex, and UK Biobank centre by deciles of age and height for the three main ethnic groups. Discussion Page 7 We found that clinical variables such as height and refractive error were associated with RPS. Reported associations of height

	we would not expect a change of pigmentation with age a priori in healthy individuals. Although age and height are associated with RPS, their effect size is small (0.02 SD change in RPS per 5-year increase in age or 5 cm increase in height), which reassures us that they likely did not affect the trends between the ethnicities. Height and age associations with RPS are of interest for future studies to validate the score in different populations.	with axial length can be a possible underpinning factor for the RPS association with height (Yin et al. 2012). Supplemental Figures We created Supplemental Figure 2 at the reviewer's request to show the associations of age and height with RPS in Asian, Black and White patients in the UK-Biobank. The y axes of supplementary Figures 1 and 2 have been updated to show the same RPS range to add clarity when comparing RPS mean differences.
-- What was the average difference between RPS of the two eyes (paired difference)?	The mean paired difference between the RPS of right and left eyes from the same participants in the UKBiobank was -1.36 with a standard deviation of 8.30. There were 30,407 participants with both right and left eyes. The one-way intraclass correlation was 0.788 (95% CI: [0.784, 0.792]). The mean difference between the RPS of the right and left eyes was very small and the ICC of 0.788 suggests a good degree of reliability within measurements within the same patient. We also showed in our separate reliability dataset (see point above), that there is an ICC greater than 0.9 for repeat imaging from the same eye, suggesting that some of the variability in RPS between	Results Retinal pigment score reliability Page 4 Among 30,407 participants in the UK Biobank that had both left and right eye fundus photos available, the mean difference was -1.36 with a standard deviation of 8.30 and a one-way ICC of 0.788 (95% CI 0.784, 0.792).

	right and left eyes in the UKBiobank may be due to the difference in appearance of the right and left eyes.	
-- Could you comment on the implications of the replication study (only 2 of 17 achieved genome wide significance)?	The EPIC-Norfolk cohort is substantially smaller than UK Biobank. However, despite reduced power in the GWAS analysis for this cohort, we observed strong replication of the genetic signals from the discovery GWAS, as evidenced by (from our results section): “The direction of effect was concordant for all 17 variants and highly correlated with estimates from the discovery analysis (Pearson’s rho = 0.986 [95% CI: 0.961, 0.995]) (Figure 4). Of the 17 variants, 15 variants were significant at $p < 0.05$, 8 remained significant after adjusting for multiple testing ($p < 0.05/17$), and 2 achieved genome-wide significance (Supplementary Table 4).” The replication GWAS does not require the same strict genome-wide significance threshold as for a discovery analysis, since a much smaller set of genetic variants (here, 17) is being analysed. The fact that 2 of these variants reached genome wide significance reflects the high	We have amended our discussion section to reflect the robustness of our GWAS replication findings as follows: Discussion Page 8 “Furthermore, despite differences in study populations and cameras, we observed robust replication for these loci in the EPIC-Norfolk cohort and a strong correlation between beta coefficients in the two cohorts.”

	strength of their association with the RPS metric.	
Overall, a potentially useful approach but the risk of overfitting, confounding and loss of too much information are concerns. Robustness studies to evaluate and better understand the finishing are important.	Thank you for your thoughtful suggestions to strengthen our manuscript.	
Reviewer #2		
This manuscript derives a new metric called retinal pigment score (RPS) that quantifies the degree of pigmentation from fundus images. It is well known that pigmentation confounds processing of fundus images, i.e., detection of vessels. Similar ideas as in this paper have previously been used to enhance fundus images (e.g., Retinal image enhancement based on color dominance of image, Scientific Reports, 2023, and refs within). However, the construction of a image derived score quantifying the pigmentation seems to be novel. 1. I was a bit disappointed with the results, RPS did not yield much biological insight or at least it wasn't highlighted in the paper.	Utilising the RPS allowed us to show several biological insights that have not been previously published, although we did not stress these in the manuscript abstract. While some of these insights are intuitive (the relationship between background retinal colour and skin/hair/eye colour) and were confirmed with both the GWAS and replication GWAS, three new genetic associations were identified that were unrelated to skin/hair/eye colour and may represent specific associations of the degree of melanin in the eye. These new associations are described in our discussion section. For instance, we are unaware of a publication showing a quantitative association	Discussion Limitations section Page 9 Secondly, RPS is currently dataset-specific, so that absolute RPS values from different cohorts cannot be directly compared if they are not fit to the same RPS scale. In this work, we elected to fit the Chinese, Australian and Tanzanian datasets to the UKB RPS scale to aid in comparison. This may be resolved with standardisation of the metric between camera types, using device-specific raw RGB values, and is subject of future work.

The main application that was highlighted was to quantify the diversity of AI training sets. Can you please elaborate, especially with respect to the fact that RPS is dataset dependent. Please also elaborate on how exactly this can be resolved with standardisation of the metric between camera types, using device-specific raw RGB values and why that wasn't done in this work.	between retinal pigmentation and refractive error or height. With respect to the fact that RPS is dataset dependent, this is because of the difficulty in standardisation across different fundus cameras. Standardisation requires modelling the proprietary non-linear function that transforms the device-specific raw RGB values that are captured with the camera sensors to the standard RGB (sRGB) values and accounting for the individual sensor properties, the illumination, and other factors that related to the way that the colour is stored. This information is not readily available and is proprietary to the cameras on which the photographs were taken. Although there have been some attempts to improve the readability of fundus photos [10.1109/TMI.2020.3043495], these change the underlying chromaticity of the images. In lieu of the original raw-RGB values, sensor information, and details about the illuminant, we plan to use post-processing techniques as a subject of future work. Although the RPS is dataset specific because of the camera types, we are reassured that the RPS of the Tanzanian, Australian and Chinese (ODIR) datasets are all in a	
---	---	--

	feasible distribution in comparison to the distributions of various UK Biobank ethnicities without correction. Currently, the RPS could be run on a training set or a test set and then used to evaluate the AI's performance across a range of RPS. One could evaluate algorithm performance across a range of RPS and report how the algorithm performs on a specific dataset. For instance, you may see that an algorithm performs well on patients who are black, but you may notice that if you bin performance by RPS, you can see that within patients who are black there is a wide degree of variation in performance depending on their RPS.	
2. In Figure 2 b the UKB scores and the scores for the Tanzanian and Australian datasets were plotted on the same plot. Is that valid since RPS is dataset dependent? Is the difference between "Black" and "Tanzanian" real?	The Tanzanian, Australian datasets were added to show that the RPS is feasible across different datasets and different demographic groups. We also added the ODIR dataset, at the request of Reviewer 1. Because the request of the reviewers was to see where these datasets fall in relation to the already published datasets, we elected to fit them with the UK Biobank RPS principal component model. This allows us to get a qualitative approximation of where the	We have modified the label naming in Figure 2b to improve the clarity.

	different datasets are compared to the UKB. We expect phenotypical differences of UKB self-reported Black and Tanzanian participants a priori because ethnicity is an imprecise measure of a person's biology. Qualitatively, we can see that the Tanzanian dataset and Black-UKB participants are similar in overall RPS distribution. This is also seen with the Chinese-UKB participants and the Chinese-ODIR dataset.	
3. The method is clearly yielding scores that are correlated with pigmentation. However, it is easy to think of other methods that does the same. Can you give some insight into why the proposed method is reasonable, e.g., why PCA instead of finding some other direction in the a,b space?	We wanted to express the degree of retinal pigment in a 1-dimensional space that still preserves the variance of the data from the three dimensional colour space of RGB data. Principal component analysis effectively reduces the dimensionality of data while retaining as much variance in the data as possible. We wanted to use a deterministic, unsupervised dimensionality reduction method that is reproducible. t-SNE and UMAP are non-deterministic and do not produce the same result each time, which means that they are not appropriate for this task. Fitting a line to the a,b colorspace data and then transforming all points onto	Results Retinal Pigment Score Page 4 The median age (IQR) was 56 years (49-63) and 92% (40,704/44,320) of participants self-described their ethnicity as white. The proportional eigenvalues of the first two dimensions were 0.899 and 0.101. The median RPS was -0.82 (-9.89, 10.39).

	the line was an option we considered, however, the major limitation with this approach is that the line would not attempt to preserve the variance from the higher dimensional space in the same way that PCA does. Because of these reasons, we feel that the PCA is appropriate. We are also reassured that the eigenvalue of the primary eigenvector from the UKB accounted for 0.899 of the explained variance (see reviewer 1 comment). The eigenvalues have been included in our main manuscript.	
4. Sometimes the manuscript lacks explanations and details, e.g., what are the number inside the parenthesis in line 106? Give more detail about the details of the QC for RPS. Were the RPS scores inverse normal transformed for GWAS? Why weren't the similar corrections used in GWAS, Phewas and statistical analysis?	The measure of spread (in parenthesis) is specified on first mention. See “The median age (IQR) was 56 years (49-63)”. Standard deviation and interquartile ranges are used as measures of spread in our work with mean and median, respectively. We have updated our multivariable linear regression model to include standardised RPS (without inverse normal transformation) to be consistent with the genetic and PheWAS analyses methods. Each unit change in the table now represents a standard deviation change in RPS. The results section text and	Results Page 5 Next, the associations were tested with multivariable linear regression adjusting for age, sex, height, self-described ethnicity, self-described hair and skin colour, Townsend index of deprivation (TDI), refractive status, and UK- Biobank assessment centre (Supplementary Table 2). The RPS was modelled as a z-score. Coefficients represent the standard deviation (SD) change in RPS per specified increase in covariates or the standardised difference between groups. The estimates in the results section “Associations of RPS with clinical variables” (Page 5),

	methods have been updated accordingly.	and Supplementary table 2 linear regression have been updated to reflect these changes.
Reviewer #3		
A well-researched and impactful work. The main objective of the paper is to develop the Retinal Pigmentation Score (RPS), a continuous measurement of retinal pigmentation derived from retinal images. The motivation behind the work is well-defined. The results justify the objective. It will help ensure the explainability and fairness of AI models.		
One concern is that in the AutoMorph pipeline, what is the justification behind using ensembled Unet architecture and not other architectures like “Hard Attention Net (HANet)” by D. Wang et. al., “Scale-space approximated convolutional neural networks (SSANet)” by K. J. Noh et. al. for Retinal Vessel Segmentation.	We used AutoMorph because it is open-source, already used and accepted in the literature for vessel segmentation and already pretrained on multiple datasets. While both SSANet and HANet are innovative techniques, neither are open-source nor do they provide weights for the models. Because of this, implementing the algorithms and training them would be difficult. First, we would have to adapt their methods into code, then we would have to find multiple robust training datasets to train the models. Additionally, HANet and SSANet improve	

	the SOTA accuracy on retinal vessel segmentation tasks by <1%. The RPS is derived from the median retinal background pixel value, and it is unlikely that a 1% change in the pixels of the vessel segmentation will affect the RPS for a fundus photo.	
The authors should present the effect of including the L vector on RPS.	To evaluate the effect of including the L vector on RPS we compared the mean paired difference between right and left eye of the same patient in the UKBiobank. We assume that the RPS should be similar. We created an RPS using only the a,b vectors and compared it to an RPS using the L,a,b vectors. The mean inpatient difference (n=30,407) between using the RPS from the a,b vector alone was -1.35 with a standard deviation of 8.30 compared to a mean of -2.40 with a standard deviation of 10.70 using the L,a,b vectors to calculate the RPS. The a,b RPS had a greater intraclass correlation coefficient (ICC) compared to the L,a,b RPS: 0.788 (95% CI: 0.784, 0.792) compared to 0.757 (95% CI: 0.753, 0.762). This demonstrates there is less inpatient variance when removing the L vector from the data, and reassuringly confirms that removing the L	Added in methods under Retinal Pigment Score Reliability: Among 30,407 participants in the UK-Biobank that had both left and right eye fundus photos available, the mean (SD) difference was -1.36 (8.30) and a one-way ICC of 0.788 (95% CI: 0.784, 0.792).. In contrast, the use of the L, a, b vectors to calculate RPS among the same group of participants yielded a mean score of -2.40 (10.70) and a one-way ICC of 0.757 (95% CI: 0.753, 0.762).

	vector makes the RPS a more precise metric.	
--	---	--

Reviewers' Comments:

Reviewer #1:

Remarks to the Author:

thank you for addressing the comments raised in the previous round.

Reviewer #2:

Remarks to the Author:

The authors have responded satisfactorily to all of my concerns.

Reviewer #3:

Remarks to the Author:

All the comments have been address by the authors.

Reviewer #4:

Remarks to the Author:

Comments related to METHODS and RESULTS:

Does the masking of the vasculature and optic nerve algorithm perform equally well in color fundus images with lighter and darker backgrounds (i.e., across races)? Some validation metrics of the masking or at least some descriptive information (e.g., the % area masked out for vasculature compared across races) would reassure the reader that the RPS is measuring the same background areas of the eye and does not include vasculature (darker) regions in the darker pigmented eyes (e.g., artificially bias the darkest RPS to even more extreme RPS because of included vasculature).

The inability to grade the colour fundus images (i.e., calculate a Retinal Pigment Score) for approximately 45% of the UK Biobank sample because they were deemed ungradable by the pipeline is problematic for deployment of the RPS algorithm in other studies. Can the authors give any insight as to why such a large proportion of the color fundus images are not gradable?

Supplementary Table 1 shows data for 44,320 persons, but the sampling “unit” on which RPS is measured is the [eye within person] unit level. How were data “combined” to the unit of “person” when a person had a gradable RPS for both their left and right eye

versus when a person had only one eye with a gradable RPS value? For example, was the RPS for persons with images of both eyes an average across the eyes?

The data in Figure 2 are summarized using quintile splits in the RPS, but data presented in Supplementary Table 1 are described using tertile groupings of RPS. Is there a reason for this difference?

THE COMMENTS BELOW ARE RELATED TO THE Multiple Linear Regression MODEL IN SUPPLEMENTARY TABLE 2:

The results in Supplementary Table 2 are interesting. However, I suspect some of them to likely be spurious. Many of the characteristics included in the multivariable model in Supplementary Table 2 are likely collinear and it is difficult to predict how they may be impacting the relationships with RPS. For example, skin colour, hair colour, race/ethnicity, and height are all likely strongly inter-correlated.

Of particular concern are the analyses related to height (and age). Height is known to be correlated with race, with taller height being more common in persons of Northern European extraction (i.e., self-identified persons of White race/ethnicity). As such, the association of height with RPS reported in the multivariable adjusted regression models in Supplementary Table 2 are problematic. Specifically, it is reported that for each 5 cm increase in height, a 0.02 standard deviation decrease in the RPS was observed ($P = 0.000000036$). While statistically significant, the size of this change is not likely of any clinical or scientific significance. Furthermore, the multivariable model assumes that the relationship (i.e., slope) of height is constant across the full range of RPS.

One approach to test the robustness of the relationship between height and RPS would be to perform stratified analyses. For example, could the authors test whether height is associated with RPS within race/ethnicity strata of White, Black, Asian, etc. one at a time? Similarly, it may be useful to test whether the age effect persists in strata by race/ethnicity. No information is given as to whether the persons who were included may have been older or younger on average in the White race/ethnicity strata, and thus the age effect could be an indirect surrogate for race, skin colour, and/or hair colour.

Another approach could be to investigate the functional form of the relationship between height (and separately age) and RPS – in regression models including splines or polynomial terms for height but not including other (collinear multivariable) terms for race/ethnicity, hair colour, and skin colour. Does height have a consistent association with RPS across the full range of RPS? Does age have a consistent association with RPS across the full range of RPS?

The associations of the RPS with race/ethnicity, skin colour, and hair colour are at least an order of magnitude greater than those for age, height, and the Townsend index of deprivation. Some nuance in how the results are reported for Supplementary Table 2 could benefit the take-away message. For example, a change in RPS of 0.02 SD, while statistically significant, is dwarfed by changes of 0.2 or even 1+ SD for other characteristics.

In summary, without providing further stratified or functional form sensitivity analyses to ensure the height and age associations with RPS are robust, I would suggest de-emphasizing these associations.

Manuscript ID: NCOMMS-23-28885A-Z

Ethnicity is not biology: retinal pigment score to evaluate biological variability from ophthalmic imaging using machine learning

The authors would like to thank the editor and the reviewers for their valuable constructive feedback and the opportunity to revise this manuscript further. We have copied the reviews on the left column, our response in the middle column, and the changes made in the manuscript on the right column. Additions to the text are displayed in text with a different colour and deletions are shown in strikethrough.

Reviewer comments	Author response	Manuscript changes
Reviewer #1		
Thank you for addressing the comments raised in the previous round. The code is well documented and the results are reproducible. The code would be a useful resource although the bulk of the code is from a previous repo (Automorph)	We thank the reviewers for their thoughtful review and time. Their suggestions have strengthened our manuscript.	
Reviewer #2		
The authors have responded satisfactorily to all of my concerns.		
Reviewer #3		
All the comments have been address by the authors.		

Reviewer #4		
Comments related to METHODS and RESULTS:		
Does the masking of the vasculature and optic nerve algorithm perform equally well in color fundus images with lighter and darker backgrounds (i.e., across races)? Some validation metrics of the masking or at least some descriptive information (e.g., the % area masked out for vasculature compared across races) would reassure the reader that the RPS is measuring the same background areas of the eye and does not include vasculature (darker) regions in the darker pigmented eyes (e.g., artificially bias the darkest RPS to even more extreme RPS because of included vasculature).	Thank you. To address this point we have conducted 2 additional analyses. Firstly, we have examined the proportion of the area masked as vasculature and as optic disc by ethnicity and show this data as suggested. For all the fundus photos in the UK Biobank we calculated the total number of pixels in each respective fundus photo and the total number of pixels from the vessel segmentation and the optic disc. We then calculated the percentage of the number of pixels in the vessel and disc segmentation relative to the overall image size. Finally, we showed the percentage of the area of the image consisting of the vessel and disc for each ethnicity along different quintiles of RPS. These results are displayed in supplementary figure 1. From the figure we can see that the percentage difference between the ethnicities is	Results Retinal pigment score (Paragraph 1) ... A total of 135,592 colour fundus photographs (67,982 right eyes, 67,610 left eyes) from 68,504 participants in the UK Biobank study were available for analysis. From these, 74,851 images (40,329 right eyes, 34,388 left eyes) from 44,320 participants (55% female) were deemed gradable by our pipeline and included in the analysis. Previous studies with manual quality grading have described a similar imbalance in laterality in this dataset.²⁵ Supplementary figure 1 shows the percentage area of the image identified as vessels and optic disc was comparable across ethnic groups. Moreover, the approximate area of the optic nerve head would fall in line with previous reports by ethnicity. Additionally, small differences in the area of the segmentation mask has a minimal effect on the RPS as shown in Supplementary Figure 2. Methods

around ~1% of the total area masked. Additionally, after breaking the RPS down by quintiles and assessing the vessel and disc segmentation size in relation to the total size of the image there are no RPS quintiles that seem to have systematically smaller or larger segmentations.

Secondly, to assess the effect of increasing or decreasing the vessel and disc segmentation area on the RPS we selected a random selection of 100 fundus photos from each of the White, Black, Asian, Chinese and Mixed cohorts of the UK Biobank. Then we randomly selected to either perform a binary erosion or binary dilation to the vessel and disc segmentation masks on each fundus photo. The erosions and dilations were run using a 3x3 kernel for 1 iteration. From the 500 color fundus photos, the mean change in the percentage of the area of the total image was 0.001 and the standard deviation was 2.383 percent. We then calculated the difference in RPS between the fundus photo with the original segmentation or the modified segmentation masks. This is

Retinal pigment score (paragraph 3)

... This new transformed vector was stored as the 1-dimensional RPS vector. Figure 1 represents a schematic of the pipeline.

Sensitivity analyses were conducted to test for RPS performance across different background colour image pigmentation by calculating the vessel and optic disc mask area from the total pixel image area by ethnicity and by RPS quintiles. The effect of increasing or decreasing the vessel and disc segmentation area on the RPS was examined with a random selection of either a binary erosion or binary dilation for 1 iteration with a 3x3 kernel to the combined vessel and disc masks for a random selection of 100 fundus images from each of the White, Black, Asian, Chinese and Mixed ethnic groups of the UK Biobank.

Supplementary material

	shown in Supplemental Figure 2. Our results show that there are minimal differences in RPS with small changes (~2%) in the percentage of the total image size that is composed of a vessel and disc segmentation. This shows that although there may be small differences in the area of segmentation masks by ethnicity, this does not have a large effect on the RPS. Therefore, a similar background area, not including either darker areas of vasculature or lighter areas of the optic nerve, is being measured for all eyes	Supplementary Figure 1: Percentage of image area covered by algorithm mask. A: percentage of vessel and optic disc mask by ethnicity and retinal pigment score (RPS) quintiles; B: percentage of vessel and optic disc mask by ethnicity; C: percentage of vessel mask by ethnicity; D: percentage of optic disc mask by ethnicity. Supplementary Figure 2: Box Plot of the difference in RPS when the algorithm was run with original vessel and disc segmentation masks compared to segmentation masks that underwent random erosions or dilations. There were 100 randomly selected fundus photos from each self-reported ethnicity in the UK Biobank cohort.
The inability to grade the colour fundus images (i.e., calculate a Retinal Pigment Score) for approximately 45% of the UK Biobank sample because they were deemed ungradable by the pipeline is problematic for deployment of the RPS algorithm in other studies. Can the authors give any insight as to why such a large proportion of the color fundus images are not gradable?	Thank you for your remark. This is explained due to the following reasons. Firstly, the UK Biobank imaging protocol entailed non-mydratiac image capture using a mydratiac fundus camera. Secondly, AutoMorph incorporates a classification model to grade image quality (https://tvst.arvojournals.org/article.aspx?articleid=2783477). The model classifies each image as good, usable, or reject quality.	Results Retinal pigment score (paragraph 2) ... The Chinese dataset was from the publicly available Ocular Disease Intelligent Recognition (ODIR) dataset.²⁷ Supplementary table 1 provides detail on the number of images analysed, included, and deemed as inadequate quality, hence excluded by the pipeline. Supplementary material

In the context of non-mydratric colour fundus photography, we only included images with good-quality for our analysis. Thirdly, this estimate expresses the percentage of ungradable images at the image level. At the patient level, the ungradable images are expected to be lower (35.3%).

A study that employed human graders to assess the quality of colour fundus photographs in the UK Biobank found that only 10.5% (14305/135592) of photos were of “Good” quality. 58.1% were “Fair”, 20.7% were “Poor” and 10.6% were “Ungradable”.

<https://www.nature.com/articles/s41433-022-02298-7>

Only roughly 11% of the photographs from the UK Biobank were of “Good” quality by human graders, which demonstrates that the high level of ungradable images is likely due to the inherent characteristics of the dataset and that our algorithm is in line with previous benchmarks set by human graders.

The rate of good-quality images in mydratric image capture or on different

We have included Supplemental Table 1 to describe the rates of ungradable images in each dataset.

	currently available datasets is expected to yield a higher proportion of gradable colour fundus images, which is seen in the other datasets used in this study. We have added the proportion of ungradable images per dataset for further clarity regarding gradeability. In the EPIC-Norfolk, Tanzanian, Chinese and Australian dataset there are lower rates of ungradable images than the UK Biobank.	
Supplementary Table 1 shows data for 44,320 persons, but the sampling “unit” on which RPS is measured is the [eye within person] unit level. How were data “combined” to the unit of “person” when a person had a gradable RPS for both their left and right eye versus when a person had only one eye with a gradable RPS value? For example, was the RPS for persons with images of both eyes an average across the eyes?	The average RPS between right and left eye was calculated to report patient-level characteristics and to investigate associations. See Results section - Association of RPS with Clinical Variables - section (page 5): “We first examined associations of mean RPS (average score between right and left eyes per participant) with sociodemographic and clinical variables.”	Supplementary table 2 legend and footnote have been modified accordingly. Supplementary information Supplementary Table 2. Baseline UK Biobank patient-level cohort characteristics by tertiles of retinal pigment score (RPS).
The data in Figure 2 are summarized using quintile splits in the RPS, but data presented in Supplementary Table 1 are described using tertile groupings of RPS. Is there a reason for this difference?	Tertiles were chosen to avoid supplementary table 1 becoming too wide. We are in agreement with your remark and we have modified the table accordingly to show characteristics by RPS quintiles in the revised version of our manuscript.	Supplementary information Supplementary table 2 has been modified to show characteristics by quintiles instead of tertiles. It has been renamed supplementary table 2 because we have added a new supplementary table 1 in response to your other concerns.

THE COMMENTS BELOW ARE RELATED TO THE Multiple Linear Regression MODEL IN SUPPLEMENTARY TABLE 2:		
The results in Supplementary Table 2 are interesting. However, I suspect some of them to likely be spurious. Many of the characteristics included in the multivariable model in Supplementary Table 2 are likely collinear and it is difficult to predict how they may be impacting the relationships with RPS. For example, skin colour, hair colour, race/ethnicity, and height are all likely strongly inter-correlated.	We conducted formal variance inflation factor testing on the final model with adjusted generalised standard error inflation factors (aGSIF). Variables included in the final model showed no strong collinearity (aGSIF for all variables less than 1.6) and were kept in the final model. (https://www.tandfonline.com/doi/epdf/10.1080/01621459.1992.10475190?needAccess=true).	Results Association of RPS with clinical variables (Paragraph 2) Coefficients represent the standard deviation (SD) change in RPS per specified increase in covariates or the standardised difference between groups. Formal variance inflation factor testing on the final model with adjusted generalised standard error inflation factors showed no strong collinearity (Supplementary table 4). Supplementary material: Supplementary table 4 has been added to show the variance inflation factor testing results.
Of particular concern are the analyses related to height (and age). Height is known to be correlated with race, with taller height being more common in persons of Northern European extraction (i.e., self-identified persons of White race/ethnicity). As such, the association of height with RPS reported in the multivariable adjusted	These were really helpful comments and suggestions. We have conducted sensitivity analyses using linear regression, stratifying by the three major ethnic groups. Recognising that the effect sizes are small, we have revised our results and discussion sections to	Results Associations of RPS with clinical variables Every 5-year rise in age was associated with a small 0.02 SD increase in RPS ($p 1.3 \times 10^{-8}$), and every 5 cm increase in height conferred a small -0.02 SD change in RPS ($p 3.6 \times 10^{-8}$). However, sensitivity analyses with stratified linear regression models across the three main

regression models in Supplementary Table 2 are problematic. Specifically, it is reported that for each 5 cm increase in height, a 0.02 standard deviation decrease in the RPS was observed (P = 0.000000036). While statistically significant, the size of this change is not likely of any clinical or scientific significance. Furthermore, the multivariable model assumes that the relationship (i.e., slope) of height is constant across the full range of RPS.	appropriately de-emphasise these specific associations. For the clinical variable of height and age, there was not a consistent directionality of association between the covariates and our stratified regression. The analysis showed that there was a positive association with age among white patients and a negative association with age among black and Asian patients. This result led us to remove a section in the discussion about the	ethnic groups showed an association in different direction for age in white ethnic groups when compared with models from black and Asian ethnic groups (supplementary table 5). The association with height remained significant and in the same direction for white and Asian ethnic group models, and was not significant for Black ethnic groups. Supplementary figure 4 shows mean RPS adjusted for sex, and UK Biobank centre by deciles of age and height for the three main ethnic groups. A non-linear association was evident for refractive status. A higher RPS was observed in people with emmetropia (0.16 [95%CI 0.11, 0.20]; p 1.1×10^{-12}), and hyperopia (0.11, [0.06, 0.15]; p 1.1×10^{-6}) when compared with people with high myopia. The most deprived TDI quintile showed a 0.06 SD increase in RPS when compared with the least deprived TDI (p for linear trend 3×10^{-4}). Townsend Deprivation Index showed, however, an association in a different direction in sensitivity analysis in the white ethnic group model when compared to the black ethnic group model (supplementary table 5).
One approach to test the robustness of the relationship between height and RPS would be to perform stratified analyses. For example, could the authors test whether height is associated with RPS within race/ethnicity strata of White, Black, Asian, etc. one at a time? Similarly, it may be useful to test whether the age effect persists in strata by race/ethnicity. No information is given as to whether the persons who were included may have been older or younger on average in the White race/ethnicity strata, and thus the age effect could be an indirect surrogate for race, skin colour, and/or hair colour.	association of age and RPS and include the sensitivity analysis in our results section. The association with height and RPS remained significant and in the same direction for white and Asian patients but was no longer significant for black patients. We added these results to the manuscript. We also added our findings with the stratified regression for Townsend Deprivation Index.	Methods Statistical analysis Linear regression models with standardised RPS (z-score) adjusting for age, sex, self-

Another approach could be to investigate the functional form of the relationship between height (and separately age) and RPS – in regression models including splines or polynomial terms for height but not including other (colinear multivariable) terms for race/ethnicity, hair colour, and skin colour. Does height have a consistent association with RPS across the full range of RPS? Does age have a consistent association with RPS across the full range of RPS?

The associations of the RPS with race/ethnicity, skin colour, and hair colour are at least an order of magnitude greater than those for age, height, and the Townsend index of deprivation. Some nuance in how the results are reported for Supplementary Table 2 could benefit the take-away message. For example, a change in RPS of 0.02 SD, while statistically significant, is dwarfed by changes of 0.2 or even 1+ SD for other characteristics.

In summary, without providing further stratified or functional form sensitivity analyses to ensure the height and age associations with RPS are robust, I would

reported ethnicity (categorised as white, black, Asian, mixed, Chinese, or other), hair colour (categorised as blonde, red, light brown, dark brown, black and other), skin colour (categorised as very fair, fair, light olive, dark olive, brown and black), spherical equivalent, height, TID (scores categorised in quintiles where a higher quintile implies a greater degree of deprivation), and UK Biobank assessment centre were used to examine associations with RPS.

Collinearity was examined using variance inflation factor testing on the final model with adjusted generalised standard error inflation factors.⁸¹ Missing data points were categorised as “Missing” within each variable. Sensitivity analyses were conducted fitting stratified linear regression models for the three main ethnic groups (white, black, and Asian ethnic groups) adjusting for age, sex, hair colour, skin colour, spherical equivalent, height, TID, and UK Biobank assessment centre.

Discussion

Paragraph 4

The first two sentences have been deleted: We found that clinical variables such as height and refractive error were associated with RPS. Reported associations of height with axial length can be a possible

suggest de-emphasizing these associations.

underpinning factor for the RPS association with height.

Reviewers' Comments:

Reviewer #4:

Remarks to the Author:

I thank the authors for their thoughtful and complete responses to comments. I have no further comments or concerns.